# MASKED GENERATIVE POLICY FOR ROBOTIC CONTROL

**Lipeng Zhuang**,*    **Shiyu Fan**,*    **Florent P. Audonnet**,    **Yingdong Ru**
**Edmond S. L. Ho**,    **Gerardo Aragon Camarasa**,    **Paul Henderson**
Department of Computer Science, University of Glasgow, Glasgow, United Kingdom

## ABSTRACT

We present Masked Generative Policy (MGP), a novel framework for visuomotor imitation learning. We represent actions as discrete tokens, and train a conditional masked transformer that generates tokens in parallel and then rapidly refines only low-confidence tokens. We further propose two new sampling paradigms: MGP-Short, which performs parallel masked generation with score-based refinement for Markovian tasks, and MGP-Long, which predicts full trajectories in a single pass and dynamically refines low-confidence action tokens based on new observations. With globally coherent prediction and robust adaptive execution capabilities, MGP-Long enables reliable control on complex and non-Markovian tasks that prior methods struggle with. Extensive evaluations on 150 robotic manipulation tasks spanning the Meta-World and LIBERO benchmarks show that MGP achieves both rapid inference and superior success rates compared to state-of-the-art diffusion and autoregressive policies. Specifically, MGP increases the average success rate by 9% across 150 tasks while cutting per-sequence inference time by up to 35×. It further improves the average success rate by 60% in dynamic and missing-observation environments, and solves two non-Markovian scenarios where other state-of-the-art methods fail.

## 1 INTRODUCTION

Enabling robots to perform complex manipulation tasks directly from high-dimensional sensory inputs, such as vision, remains a challenge in robotics and artificial intelligence. Imitation learning has emerged as a promising and data-efficient paradigm to tackle this challenge, bypassing the complex modeling process of reinforcement learning by directly leveraging human demonstrations (Zare et al., 2024). Moving beyond simple state-action mapping, recent advancements in learning visuomotor policies have formulated the problem as training a generative model over action sequences conditioned on observations. These approaches typically use either (1) diffusion policies, which generate behavior via a conditional denoising process in robot action space (Janner et al., 2022; Chi et al., 2023; Ze et al., 2024), or (2) autoregressive policies, which treat actions as discrete tokens and model these tokens with a GPT-like transformer (Mete et al., 2024).

State-of-the-art generative policies face inherent trade-offs that limit their use in *closed-loop, real-time robotic control*. Diffusion-based methods require multiple denoising steps per action, making them computationally slow. Consistency Policy (Prasad et al., 2024) and Flow Policy (Zhang et al., 2025a) have aimed to speed up sampling but they either require additional distillation or compromise sample quality. Autoregressive policies, on the other hand, sample one token per forward pass; therefore, latency scales with sequence length. Moreover, without memory, these policies lack robustness to missing observations and fail in non-Markovian tasks.

Motivated by these limitations, our goal is to develop *generative policies that simultaneously address the inference time bottlenecks of iterative sampling and the robustness challenges inherent in non-Markovian, long-horizon manipulation tasks*. For this, we introduce **Masked Generative Policy** (MGP), which achieves low inference latency and high task success rates while supporting rapid plan edits during execution. MGP adapts masked generative transformers (Chang et al., 2022) to model latent action representations. Actions are encoded as discrete token sequences via vector

---

*Equal contribution. {`Lipeng.Zhuang, Shiyu.Fan`}@glasgow.ac.uk

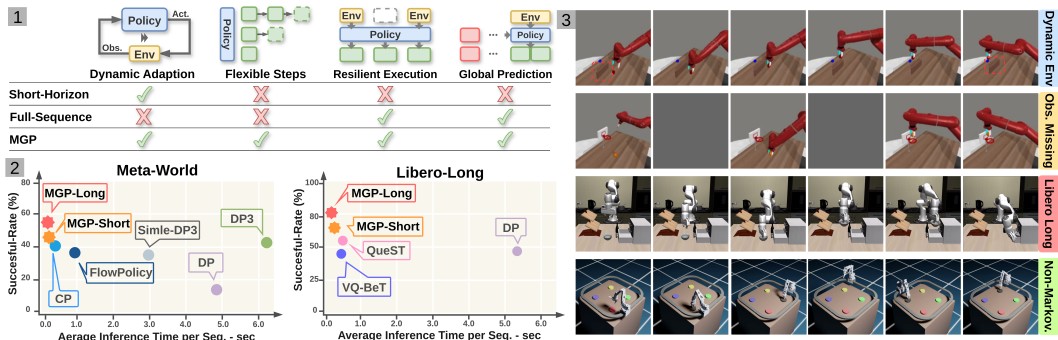

Figure 1: **Overview of MGP.** (1) Four properties of MGP: dynamic adaptation, flexible replanning steps, resilient execution, and global-coherent prediction. (2) Versus prior SOTA, MGP is both **faster** (lower per-sequence inference time) and **better** (higher success). (3) MGP also excels on several challenging settings: dynamic environments, observation-missing environments, and non-Markovian, long-horizon tasks.

quantization and a conditional masked transformer reconstructs complete sequences from partial masks conditioned on observations.

We also propose two novel sampling paradigms *MGP-Short* and *MGP-Long*. *MGP-Short* samples action tokens in parallel with few iterative refinements, achieving high success rates and minimal latency for closed-loop control on Markovian tasks. *MGP-Long* integrates a novel **Adaptive Token Refinement (ATR)** strategy, predicting global trajectories and iteratively refining actions on-the-fly using updated observations. During refinement, informed by the new observations, our *Posterior-Confidence Estimation* selectively masks and corrects unexecuted tokens with low likelihood. As a result, MGT-Long enables globally coherent predictions over long horizons while still being responsive enough to be used in closed-loop robotic control for non-Markovian tasks.

We show in Section 4 that MGP-Short and MGP-Long achieve state-of-the-art performance on 150 manipulation tasks across three standard benchmarks under single- and multi-task training. Moreover, MGP-Long demonstrates strong adaptation and robust planning strengths in dynamic and missing-observation environments, and further excels on non-Markovian scenarios. This is due to MGP-Long's globally-coherent predictions conditioned on executed action tokens, dynamic confidence-driven token updates, resilient execution under missing observations, and flexible and efficient inference via variable step sizes and targeted edits rather than full regeneration. Together, these results establish MGP as a *fast*, *accurate*, and *adaptive* new paradigm that enables *globally-coherent predictions* for visuo-motor policy learning. Our main contributions are threefold:

- We introduce Masked Generative Policy (MGP), the first masked generative framework for robot imitation learning, which eliminates diffusion models' inference bottlenecks and autoregressive models' sequential constraints.

- For Markovian tasks, we develop MGP-Short, which adapts the masked generative transformer for short-horizon sampling. We show that MGP-Short achieves better success rates on standard benchmarks while substantially reducing inference time.

- We propose MGP-Long to enable globally-coherent predictions over long horizons, dynamic adaptation, resilient execution under partial observability, and efficient, flexible execution. As a result, MGP-Long achieves state-of-the-art results on *dynamic*, *observation missing* and *non-Markovian*, long-duration environments.

## 2 RELATED WORK

**Diffusion Models for Robotic Manipulation.** Imitation learning enables an agent to acquire an expert policy without explicit rewards by training on expert (Schaal, 1996; Ho & Ermon, 2016). This is carried out typically via behavior cloning (Pomerleau, 1988; Torabi et al., 2018; Florence et al., 2021) or inverse reinforcement learning (Ng & Russell, 2000; Peng et al., 2018; Fu et al., 2018). With the advent of deep learning, directly learning expert behaviors through time-series models has gradually become the dominant paradigm (Zhao et al., 2023; Florence et al., 2019).

Recent works (Chi et al., 2023; Xian et al., 2023) have shown that diffusion models (Ho et al., 2020) can learn visuomotor policies by treating action synthesis as a conditional denoising process. Similarly, Ze et al. (2024) and Ke et al. (2024) have proposed to enhance the performance of diffusion-based policies by incorporating multimodal information such as point clouds and text instructions. Meanwhile, diffusion models have also been widely applied to trajectory planning (Zheng et al., 2025; Xiao et al., 2023). Due to the long trajectory of denoising, several studies have attempted to improve its sampling efficiency, for example, FlowPolicy (Zhang et al., 2025a) and Consistency Policy (Prasad et al., 2024). However, these methods either compromise sampling quality or require additional distillation.

**Autoregressive Models for Robotic Manipulation**   The stepwise prediction of actions in autoregressive models aligns naturally with the sequential operation of robotic task execution (Zhang et al., 2025b). Therefore, researchers have explored a variety of autoregressive models for imitation learning (Sun et al., 2017; Xie et al., 2020), with transformer-based architectures gradually emerging as the dominant approach due to their strong sequence modeling capacity (Cui et al., 2023; Brohan et al., 2023; Chebotar et al., 2023; Zhao et al., 2023). More recently, large language models have extended transformer-based architectures and introduced the next-token prediction paradigm into this field, where GPT-like pipelines encode actions into discrete embeddings via a tokenizer, and the transformer models the correspondence between these embeddings and observations (Shafiullah et al., 2022). PRISE (Zheng et al., 2024),VQ-BeT (Lee et al., 2024) and QueST (Mete et al., 2024) have achieved improved task success rates and generalization by employing novel discrete representations. Other work like Chain-of-Action (Pan et al., 2024) further extends autoregressive approaches by decoding trajectories in reverse from a goal keyframe to mitigate compounding error. However, autoregressive policies have inherent structural limitations: (i) next-token prediction leads to multi-step iterations in long-horizon action generation, and (ii) immutability of prefixes—any edit needs regenerating all subsequent tokens.

**Masked Generative Transformers**   The masking mechanism was initially employed as a regularization technique in deep learning (Vincent et al., 2008; Pathak et al., 2016; Bao et al., 2022; He et al., 2022). Building on this foundation, MaskGIT (Chang et al., 2022) and MUSE (Chang et al., 2023) have established masked token modeling as a scalable, high-fidelity paradigm for image and text-to-image synthesis, while StyleDrop (Sohn et al., 2023) demonstrates controllable, personalized generation via token-level conditioning. This paradigm has since extended beyond images. For example, MMM (Pinyoanuntapong et al., 2024) and MoMask (Guo et al., 2024) have applied masked generative modeling to long-horizon human motion.

## 3   METHODS

MGP is a stochastic policy that at time $t$ samples a sequence of future actions $\mathbf{a}_t \in \mathbb{R}^{T_f \times j}$ (where $T_f$ is the length of future predicted actions) given a conditioning $c_t$ including $T_p$ past actions, robot states $s_t$, and visual observations $O_t$. Here, $j$ denotes the dimensionality of the robot action space, typically end-effector (EE) absolute position, rotation, and gripper state. In our setting, $O_t$ represents an input observation at a given $t$, which may comprise RGB images, $o_{image} \in \mathbb{R}^{T_p \times w \times h \times 3}$, depth images $o_{depth} \in \mathbb{R}^{T_p \times w \times h \times 1}$ and/or point clouds $o_{pc} \in \mathbb{R}^{T_p \times 1024 \times 3}$.

MGP is trained by imitation learning on a dataset of expert demonstrations. We first train an Action Tokenizer that learns a discrete representation of robot action sequences (Sec. 3.1). A Masked Generative Transformer (MGT) then learns to reconstruct masked sequences of action tokens (Sec. 3.2), allowing the generation of future actions conditioned on observations. Building upon MGT Chang et al. (2022), we introduce two sampling paradigms, MGP-Short for Markovian (Sec. 3.3) and MGP-Long for Non-Markovian tasks (Sec. 3.4).

### 3.1   ACTION TOKENIZER

We use a VQ-VAE (Van Den Oord et al., 2017) to obtain a discrete representation of actions. Specifically, the VQ-VAE takes a sequence of continuous-valued actions $\mathbf{a}$ as input and maps this to a shorter sequence of discrete action tokens, which can be reconstructed to the corresponding actions (Fig. 2-1). This creates a discrete latent space for the MGT to operate over.

**Action Tokenization.**   The tokenizer is designed to discretize actions. It encodes the input actions, $\mathbf{a} \in \mathbb{R}^{T \times j}$, into a latent representation, $\hat{\mathbf{y}} \in \mathbb{R}^{N \times d}$, through two residual 1D CNN blocks, where $d$

represents the codebook dimension and $N$ is the dimension over time. For each $d$-dimensional vector, the closest token embedding, $\mathbf{y} \in \mathbb{R}^{N \times d}$, is then retrieved from a learnable codebook, $\mathcal{K} = \{\mathbf{k}\}_{k=0}^{|\mathcal{K}|-1} \in \mathbb{R}^d$, for decoding. During decoding, the decoder employs symmetric upsampling Conv1D blocks to reconstruct the action sequence $\hat{\mathbf{a}}$ of length $T$.

**Training.** The training objectives of the model follow the VQ-VAE (Van Den Oord et al., 2017), which includes reconstruction loss and commitment loss:

$$\mathcal{L}_{\text{VQ}} = \lambda_{\text{rec}} \|\mathbf{a} - \hat{\mathbf{a}}\|_1 + \beta \|\hat{\mathbf{y}} - \text{sg}[\mathbf{y}]\|_2^2 \tag{1}$$

where $\text{sg}[\cdot]$ is the stop-gradient operator, $\lambda_{\text{rec}}$ is 1 and $\beta$ is 0.02. We use exponential moving averages (EMA) for codebook updates, and resetting of inactive codewords to guarantee codebook usage (Chang et al., 2022). Once trained, the weights of the VQ-VAE are frozen. During the training of the subsequent MGT, the VQ-VAE is employed to encode the actions from the training data and to decode the tokens generated by the MGT. In the inference phase, only the VQ-decoder is used to reconstruct actions from the tokens sampled by the MGT.

## 3.2 Masked Generative Transformer

MGT is required to generate $N$ future action tokens sequence $\mathbf{y}^{0:N}=[y_n]_{n=0}^N$ conditioned on the observed inputs $O_t$ and the robot's historical states $s_t$ from unknown tokens by a few gradual refinement steps. Initially, a special learnable token $[\text{MASK}]$ is introduced to represent unknown tokens. In addition, [END] and [PAD] tokens are used to indicate the termination and padding of a token sequence, respectively (Pinyoanuntapong et al., 2024). The pipeline is illustrated in Fig. 2-2.

**Structure.** MGT samples all tokens in parallel, in contrast to the autoregressive models like GPT. The MGT consists of two components: (1) a perception encoder and (2) an encoder-only transformer. The perception encoder is used to get the observation conditions' embeddings. After obtaining an $O_t$ and $s_t$, the perception encoder encodes them through MLP layers into two feature sets, which are concatenated along the hidden dimension to construct the conditioning for the transformer. The transformer itself is composed of 2 cross-attention layers followed by 2 self-attention layers. In the cross-attention layers, the blocks compute the cross-attention map between the observation embedding and token embeddings that obtained through VQ-VAE tokenization of the ground-truth actions. The outputs of the transformer are the logits of predicted tokens.

**Training.** A subset of tokens is randomly masked and replaced with a $[\text{MASK}]$ symbol. The remaining tokens are randomly perturbed at a fixed ratio by substituting them with alternative indices. The embedding of the observations $O_t$ and $s_t$ are fed into the MGT together with the corrupted action tokens. $\mathbf{y}_{\overline{M}}$, and MGT predicts the probabilities of tokens $p(y_n|\mathbf{y}_{\overline{M}}, c)$. It is trained to minimize the negative log-likelihood, i.e. the cross-entropy between the ground-truth and predicted tokens:

$$\mathcal{L}_{\text{MGT}} = - \mathop{\mathbb{E}}_{\mathbf{y} \in \mathcal{K}} \left[ \sum_{\forall n \in [0,N]} \log p(y_n \mid \mathbf{y}_{\overline{M}}, \mathbf{c}) \right]. \tag{2}$$

## 3.3 Short horizon Mask-and-Refine Sampling (MGP-Short)

For simple tasks, decision-making does not rely on historical state information—they can be treated as a Markov decision process (MDP), and thus there is no need to explicitly model long-term state dependencies. For this setting, we propose *MGP-Short*, a basic closed-loop inference scheme that accelerates sampling while enabling dynamic adjustments during generation based on the current state (See Fig. 2-3).

*MGP-Short* only samples the tokens conditioned on the current observation $c_t$ obtained from the perception encoder. This procedure involves 2 iterations. For the first iteration, at time step $t$, the masked token sequence, together with $c_t$, is fed into the transformer to generate a first estimate of the token logits in parallel. The $n$-th token index is calculated from the raw logits $e_n$ by the Gumbel-Max trick (Huijben et al., 2022):

$$y = \arg\max_n \left( \frac{e_n}{\tau} + g_n \right), \quad g_n = -\log\big(-\log(u_n)\big), \quad u_n \sim \text{Uniform}(0,1). \tag{3}$$

where $\tau$ adjusts the temperature. Gumbel-Max (6) sampling preserves modes while increasing diversity compared with softmax sampling. Following sampling, the normalized probabilities of logits are ranked as confidence scores, and tokens with the lowest confidence scores are re-masked (5) and regenerated during the second iteration. Parallel token generation, combined with a log-likelihood–based masking strategy enable high-quality token generation within only a few iterations.

### 3.4 LONG HORIZON MASK-AND-REFINE SAMPLING (MGP-LONG)

To address limitations of existing methods in terms of efficiency and long-horizon tasks, we extend MGP to long-horizon action generation, *MGP-Long*. Through Adaptive-Token-Refinement (ATR), *MGP-Long* samples an initial nearly full episode action sequence given the initial observation and begins executing this; then during the rollout, it progressively refines the yet-to-be-executed action tokens as new observations arrive, while retaining executed actions (Fig. 3). Specifically, *posterior-confidence estimation* enables the model to continuously update a confidence score using current observations and historic states.

During inference, *MGP-Long* predicts action tokens in parallel, similarly to *MGP-Short*. At the beginning of a task, based on the initial observation condition $c_0$, *MGP-Long* infers the conditional probability $p(\mathbf{y}_0^{0:N}|c_0)$ of token and samples the complete sequence of tokens $\mathbf{y}_0^{0:N}$ covering the entire task horizon as its initial prediction. The robot can then be directed to execute tokens with a freely adjustable step length. After the $i$-th execution of $n$ tokens, with token $\mathbf{y}_{i-1}^{0:n}$ executed and $\mathbf{y}_{i-1}^{n:N}$ still pending, an updated observation $c_i$ is received from the environment and the subsequent tokens are updated in response to the new observation.

**Posterior-confidence estimation.** To ensure efficient utilization of generated tokens while maintaining global-coherence, we introduce *Posterior-confidence estimation*, a novel masking and refinement mechanism for the Adaptive-Token-Refinement. Suppose that the hidden state for this process $\mathcal{H}_i$ is provided by the tokens that have already been executed. The transformer will compute the new probability of the previously sampled result under the new observation $c_i$ as a preliminary confidence score as:

$$\mathcal{S}(\mathbf{y}_{i-1}^{0:N}) \sim p(\mathbf{y}_{i-1}^{0:N}|c_i, \mathcal{H}_{i-1})) = \text{softmax}(\mathbf{e}^{0:N}); \tag{4}$$

this is analogous to finding an updated Bayesian posterior predictive distribution given the new observation. Since the historical tokens have already been executed, their scores should be excluded from computation. Only the scores of tokens from $n$ to $N$ are normalized, and tokens with low scores are re-masked based on the ranking.

$$\mathbf{y}_{i-1\overline{M}}^{n:N} \leftarrow \text{MASK}(\mathbf{y}_{i-1}^{n:N}, S(\mathbf{y}_{i-1}^{n:N})) \tag{5}$$

The re-masked tokens, together with the previously executed historical tokens, are then fed back into the transformer for refinement, yielding the updated token logits:

$$\mathbf{y}_i^{n:N} = \text{GumbelMax}(p(\mathbf{y}_i^{n:N}|\mathbf{y}_{i-1\overline{M}}^{0:N}, c_i, \mathcal{H}_{i-1}). \tag{6}$$

Afterward, the token indices are sampled by the Gumbel-Max trick (6) and then decoded into actions by VQ-VAE. Since historical tokens are retained in the recursive generation process and incorporated into the refinement of subsequent tokens, the model is able to preserve memory of previously executed steps.

## 4 EXPERIMENTS AND RESULTS

In this section, we first evaluate on three standard benchmarks, Meta-World (Yu et al., 2020), LIBERO-90 (Liu et al., 2023), and LIBERO-Long, covering both short- and long-duration tasks under single-task and multi-task training (Section 4.1). Beyond these, we have designed three challenging evaluation environments that are important for long-horizon control and have repeatedly challenged prior methods: observation missing environments (Section 4.2); dynamic environments with moving objects/obstacles (Section 4.3); and two long-duration non-Markovian tasks (Section 4.4). Finally, we conduct four ablation studies (Section 4.5).

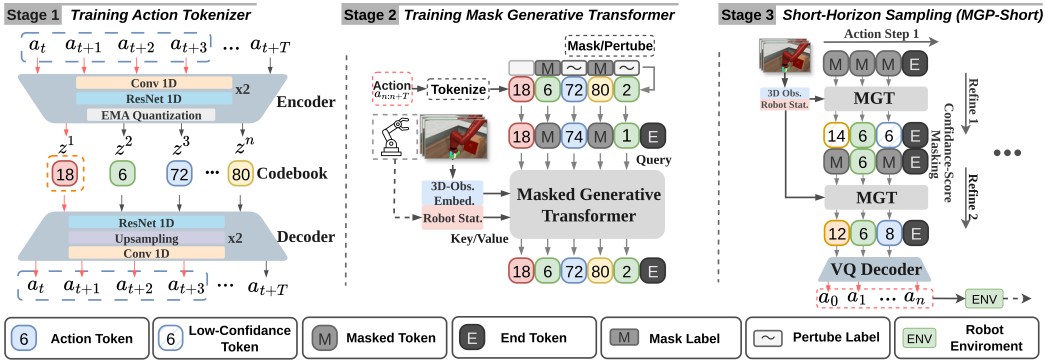

Figure 2: Left: Training Stage 1 - Action Tokenizer and Middle: Training Stage 2 - Masked Generative Transformer and Right: Short-horizon sampling (MGP-Short)

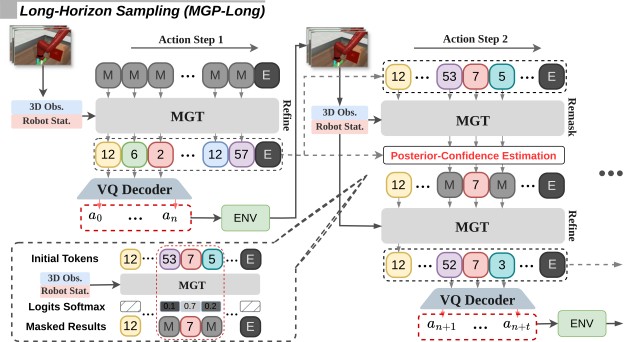

Figure 3: Long-horizon sampling (MGP-Long) through Adaptive Token Refinement (ATR).

**Model variants.** In addition to MGP-Short and MGP-Long, we include two ablations of MGP-Long: (1) *Full-Horizon MGP* (MGP–Full seq.), which generates the full trajectory once without dynamic online adaptation; and (2) *MGP-Long without score-based mask* (MGP-w/o SM), which replans by masking all remaining tokens whenever new observations arrive and regenerating the remainder from scratch. These two ablations of MGP-Long separate: (1) the benefit of planning the whole sequence, (2) the need for targeted edits for efficiency and stability, and (3) the advantage of preserving unmasked tokens to maintain global consistency across the trajectory.

**Baselines.** We evaluate against ten baselines: diffusion methods-Diffusion Policy(Chi et al., 2023), 3D Diffusion Policy (include simple DP3)(Ze et al., 2024)—two accelerated diffusion methods (Consistency Policy(Prasad et al., 2024) and FlowPolicy(Zhang et al., 2025a)) targeting faster sampling; and autoregressive methods—QueST(Mete et al., 2024), VQ-BeT(Lee et al., 2024), PRISE(Zheng et al., 2024), ACT(Zhao et al., 2023), and ResNet-T(Liu et al., 2023). Beyond the standard baselines, we also evaluate *Full-Horizon DP3* (DP3-Full Seq.), a diffusion baseline that plans the entire action sequence from the initial observation and executes it open-loop (no replanning), similar to MGP-Full Seq.

All comparisons use identical visual encoders and demonstration numbers, ensuring that improvements are from MGP-Short and MGP-Long. Implementation details of standard benchmarks are shown in Appendix Sec. A.3.

## 4.1 STANDARD BENCHMARKS

**Separate Training on Meta-World.** We run single-task experiments on all 50 Meta-World tasks spanning Easy to Very Hard (Seo et al., 2023). We use only short horizon methods on the Easy and Medium tasks, but long horizon on Hard and Very Hard tasks that benefit from a longer planning window. Ten expert demonstrations per task are generated using Meta-World heuristic policies. We

Table 2: Success Rate (SR) and Inference Time per step (Inf.T) of Single-Task Training on Meta-World. 'Inf. T per step' represents inference time per step (ms/step). 'Inf. T per seq.' represents inference time per sequence (ms/sequence).

| Methods | Meta-World | | | | | | |
|---|---|---|---|---|---|---|---|
| | Easy (28) | Medium (11) | Hard (5) | Very Hard (5) | Avg. SR | Avg. Inf. T per step | Avg. Inf. T per seq. |
| DP | 0.836 | 0.311 | 0.108 | 0.266 | 0.380 | 106 | 4750 |
| Simple-DP3 | 0.868 | 0.420 | 0.387 | 0.350 | 0.506 | 63 | 2830 |
| DP3 | 0.909 | 0.616 | 0.380 | 0.490 | 0.599 | 145 | 6557 |
| CP | 0.912 | 0.627 | 0.400 | 0.510 | 0.612 | 5 | 230 |
| FlowPolicy | 0.902 | 0.630 | 0.392 | 0.360 | 0.571 | 19 | 850 |
| **MGP-Short (ours)** | **0.920** | **0.650** | **0.440** | **0.538** | **0.637** | **3** | **135** |

evaluate 20 episodes every 1000 iterations using 3 seeds. We compute the average of top-5 success rates per task, per-step inference latency (ms) and per-sequence inference latency (ms).

In Tab. 2, we see that MGP-Short achieves state-of-the-art performance across the 50 Meta-World tasks for all difficulty levels, with an overall average success rate of 0.637, 3.8% higher than DP3 and 6.6% higher than FlowPolicy. MGP-Short takes just 3 ms per step, which is $49\times$ faster than DP3 (145 ms). Notably, it is also faster than CP and FlowPolicy, without their additional distillation or approximations.

In Tab. 1, we show results of long-horizon methods on ten Hard and Very Hard tasks. MGP-Long improves over MGP-Short by about 10% on Hard and approximately 5% on Very Hard, and for DP3, by 16% and about 10%, respectively, while reducing sequence-level latency from 135 ms (MGP-Short) and 6,557 ms (DP3) to 80 ms.

Compared with other long-horizon baselines, MGP-Long outperforms DP3-FullSeq by 35.2% and 23.6%, respectively. Relative to the two ablations, MGP-FullSeq and MGP-w/o-SM, MGP-Long further raises success by 22.3% and 2.2%, respectively, demonstrating the benefit of dynamic, targeted refinements, and the advantage of preserving high-confidence subsequences as anchors while updating only uncertain tokens. Detailed results are shown in Appendix Sec. C.1.1.

Table 1: Success Rate of Long-horizon methods under Single-Task Training on Meta-World.

| Methods | Hard (5) | Very Hard (5) | Avg. SR |
|---|---|---|---|
| DP3-Full Seq. | 0.188 | 0.350 | 0.270 |
| MGP-Full Seq. | 0.294 | 0.386 | 0.340 |
| MGP-w/o SM | 0.510 | 0.572 | 0.541 |
| **MGP-Long (ours)** | **0.540** | **0.586** | **0.563** |

**Multi-task training on LIBERO-90**

We evaluate the multi-task imitation-learning capability of MGP-Short on LIBERO-90, a suite of 90 language-conditioned manipulation tasks. We focus on MGP-Short here because LIBERO-90 features short-horizon tasks. Following QueST, we use 50 demonstrations per task and train a single multi-task policy. Each task is evaluated on 50 held-out episodes from a predefined set.

Results are shown in Tab. 3. MGP-Short achieves an average success rate of 0.889, comparable to QueST (0.886) and exceeding all other baselines including 13.5% higher than DP and 7.6% higher than VQ-BeT. Crucially, it achieves this accuracy with sub-stantially lower per-step inference latency, reducing

Table 3: Success Rate of Multi-Task Training on Libero90 and Libero-Long.

| Methods | Libero90 | Libero-Long |
|---|---|---|
| ResNet-T | 0.844 | 0.441 |
| ACT | 0.508 | - |
| DP | 0.754 | 0.501 |
| PRISE | 0.544 | - |
| VQ-BeT | 0.813 | 0.593 |
| Quest | 0.886 | 0.680 |
| **MGP-Short (ours)** | **0.889** | **0.770** |
| MGP-w/o SM | - | 0.805 |
| **MGP-Long (ours)** | - | **0.820** |

time from 17 ms to 5 ms relative to QueST, because it predicts tokens in parallel and uses only one or two selective refinement passes rather than token-by-token decoding. Detailed results of 90 tasks are shown in Appendix Sec. C.1.2.

**Multi-task training on Long-duration Libero-Long**

We evaluate MGP-Short and MGP-Long on long-duration manipulation using LIBERO-Long in the multitask setting. We follow the evaluation protocol and training and inference as Section 4.1.As

Table 4: Success Rate of Single-Task Training on Observation missing and Dynamic environments.

| Methods | *Obs. Missing* Meta-World | | | *Dynamic* Meta-World | | | | | |
|---|---|---|---|---|---|---|---|---|---|
| | Hard(5) | Very Hard(5) | Avg.SR | Basketball | Pick place wall(W) | Pick place wall(T) | Push | Push wall(T) | Avg.SR |
| DP3 | 0.160 | 0.240 | 0.200 | 0.91 | 0.35 | 0.07 | 0.20 | 0.27 | 0.360 |
| **MGP-Short (ours)** | 0.172 | 0.238 | 0.205 | 0.92 | **0.40** | **0.15** | 0.23 | 0.45 | 0.430 |
| DP3-Full Seq. | 0.188 | 0.350 | 0.269 | 0.02 | 0.10 | 0.00 | 0.05 | 0.03 | 0.040 |
| MGP-Full Horizon | 0.294 | 0.386 | 0.340 | 0.05 | 0.13 | 0.05 | 0.09 | 0.20 | 0.100 |
| MGP-w/o SM | 0.416 | 0.538 | 0.477 | 1 | 0.25 | 0.12 | 0.20 | 0.50 | 0.396 |
| **MGP-Long (ours)** | **0.484** | **0.566** | **0.525** | **1** | 0.3 | 0.13 | **0.25** | **0.50** | **0.436** |

shown in Tab. 3, MGP-Short achieves an average success rate of 77%, outperforming QueST by 9% and reducing the per-step inference latency from 16.3 ms per step to 4.5 ms. These gains indicate that the masked-generative framework is well-suited to long-duration tasks because it mitigates the sequential bottleneck of autoregressive decoding and reduces compounding errors by capturing longer-range dependencies. MGP-Long further improves performance, reaching 82.0% average success, increasing success by 5% over MGP-Short. The ablation MGP-w/o-SM reaches 80.5%, lower than MGP-Long by 1.5%, confirming the benefit of score-based confidence masking. In addition, MGP-Long achieves the fastest sequence-level inference time, dropping from 225 ms to 78 ms compared with MGP-Short. Detailed results are shown in Appendix Sec C.1.3.

**Model Size and Speed**

Our two-stage system is lightweight and fast: 7M parameters (**37×** fewer than DP3 Ze et al. (2024)'s 262M); training for 2000 epochs takes 55 minutes (10 minutes for Stage one and 45 minutes for Stage two) vs. 3 hours on the same RTX 4090 setup. See Appendix Sec. C.6 for details.

## 4.2 ROBUSTNESS TO MISSING OBSERVATIONS

To evaluate robust action continuation under partial observability, we test with observation dropouts. At each control cycle during inference, the current observation is withheld with some probability, forcing the policy to execute actions without fresh sensory input. We use 10 Meta-World Hard/Very Hard tasks; training follows Section 4.1. For short-horizon baselines (DP3, MGP-Short), when a dropout occurs, the controller holds position (zero-action/hold) until the next observation arrives, then generates a new action clip. In contrast, MGP-Long continues executing its *already planned* actions from the initial full-horizon proposal and previous steps refinement and skips the *score-based masking scheme* while observations are missing until sensing resumes.

Results in Tab. 4 (left) show MGP-Long achieves an average success of 0.484 on Hard and 0.566 on Very Hard, gains of roughly 22%–31% over the short-horizon methods. Short-horizon policies failed because dropouts yield static, out-of-distribution point clouds with little motion signal. By contrast, MGP-Long follows a full-horizon planning and retains the high-confidence future tokens as anchors, making it resilient to partial observability than windowed clip decoding. We further evaluate MGP-Long as the observation-drop probability increases from 0.35 to 0.70 and find that it remains robust to missing observations. Detailed results are provided in Appendix Sec. C.2.

## 4.3 DYNAMIC ENVIRONMENTS

We construct dynamic variants of five Meta-World tasks, in which key scene elements move continuously during execution: a translating hoop in *Basketball*, a moving wall in *Pick Place Wall–Wall*, a drifting goal in *Pick Place Wall–Target*, and moving targets in *Push* and *Push (Wall)*.

Results are shown in Tab. 4 (right). Existing full-sequence methods tend to fail: DP3-FullSeq achieves an average success of 0.04 and MGP-FullSeq 0.10, indicating that one-shot, open-loop rollouts cannot cope with continual scene changes. In contrast, MGP-Long delivers the best performance with 0.436 average success. MGP-Long achieves marginally better performance than MGP-Short (0.43; +0.6%), and both clearly surpass DP3 (0.36; +7.6%). These results demonstrate MGP-Long's dynamic adaptation: it maintains a globally coherent trajectory while performing quick in-place refinements to track moving targets and avoid shifting obstacles. Detailed environments setup and qualitative results are in Appendix Sec A.4 and Sec. C.3.

## 4.4 Non-Markovian Environments

To assess MGP-Long's globally-coherent predictions, we design two non-Markovian tasks. Both simulate an 80x80cm tabletop with a rail forming a loop along the perimeter; a LeRobot SO101[1] is mounted on a carriage that follows the rail. Push-buttons are placed near the four table corners. In **Button Press On/Off**, each button is lit red, green, blue or yellow. For each episode, colors are randomly assigned to corners and all lights are on. The robot must press buttons in a fixed color order such as red $\rightarrow$ green $\rightarrow$ blue $\rightarrow$ yellow, independently of the button's location. The scene looks identical after any button press, so progress is unobservable from a single frame. In **Button Press Color Change**, each button cycles through five states when pressed: yellow $\rightarrow$ red $\rightarrow$ green $\rightarrow$ blue $\rightarrow$ off. The four buttons start with different colors. The task is to press buttons in the order of their initial colors. Thus multiple buttons may display the same color during execution (e.g. after pressing the green button, there will be two blue buttons), and identical frames can correspond to different stages. See Appendix Sec. A.5 for detailed descriptions and dataset collection procedures. To evaluate, we use 20 rollouts and report the success rate.

In Tab. 5, MGP-Long achieves the highest success on both non-Markovian button tasks. In both tasks, short-horizon methods (DP3, QueST, MGP-Short) often fail to complete the sequence, because identical frames provide no observable progress signal. By contrast, MGP-Long maintains a full-horizon plan and performs confidence-guided, in-place refinements, enabling it to complete the prescribed color order. Qualitative results see Appendix Sec. C.4

Table 5: Success Rate on two Non-Markovian environments.

| Method | Button Press On/Off | Button Press Change Color |
|---|---|---|
| DP3 | 0.00 | 0.00 |
| QueST | 0.00 | 0.00 |
| **MGP-Short (ours)** | 0.00 | 0.00 |
| **MGP-Long (ours)** | **1.00** | **1.00** |

## 4.5 Ablation study and Hyperparameter experiments

We conduct seven ablation and hyperparameter experiments. Full results are given in Appendix B.

**Refine steps for MGP-Short:** On five hard Meta-world tasks, increasing mask-refine iterations from $r = 1$ to $r = 2$ yields a gain of 14.3%, validating the *score-based masking scheme*, while $r = 3$ does not offer a significant benefit and adds latency; therefore, we adopt $r = 2$ as default.

**Codebook size for MGP-Short:** We explore the influence of codebook size on five Meta-World Very Hard tasks, training separate models with codebook sizes of 512, 1024, and 2048. The average success rates are 0.534, 0.538, and 0.522, respectively. The gaps are small (1.6%) and show no consistent trend, indicating minimal sensitivity to codebook size.

**Discretization granularity for MGP-Short:** We test with 2 actions/token and 8 actions/token on the five Meta-World Very Hard tasks. Averaged over tasks, 4 actions/token achieves the highest success rate (0.538), outperforming 2 actions/token (0.526, +1.2%) and 8 actions/token (0.514, +2.4%). The gaps are modest, indicating limited sensitivity to granularity.

**Refine steps for MGP-Long in challenging environment:** We evaluated $r \in \{1, 2, 3\}$ on five dynamic tasks of MGP-Long. Increasing the refinement steps from $r = 1$ to $r = 2$ raises the success rate by +5.2%, indicating that the score-based masking scheme is helpful in more challenging settings. Increasing further from $r = 2$ to $r = 3$ yields $< 1\%$ improvement, while incurring higher inference cost. We therefore set $r = 2$ for MGP-Long in the main experiments.

**Mask ratio for MGP-Long:** We vary what proportion of low confidence tokens are resampled. When replanning we rank the unexecuted tokens by posterior confidence and in our standard setting mask the bottom 70% for one refinement pass. Changing between 50%, 70%, and 85% shows that 70% yields the best average success across five Meta-World Very Hard tasks. A 50% ratio underperforms because many low-confidence tokens remain unedited and can also interfere with accurate regeneration of the masked ones; 85% is comparable to 70% on average.

**Scoring policy for MGP-Long:** At each replanning step $t$, we ablate three confidence–update schemes to choose which future tokens to re-mask first prior to refinement: (1) *Random*: Use random scores to mask the rest of tokens, (2) *Score Reuse*: Reuse the previous confidences from last

---

[1]https://github.com/TheRobotStudio/SO-ARM100

steps to select the lowest-confidence tokens; and, (3) *ATR*: Recompute current confidences from the latest observations using the masked generative transformer first, then select only the low-confidence tokens. *ATR* attains the highest success, 10.68% over *Random* and 5.53% over *Score Reuse*.

**Varying execution step for MGP-Long:** On five hard Meta-World tasks, we evaluate MGP-Long across execution step sizes of 4, 12, and 36. A step size of 12 yields the highest success (54%), while steps of 4 and 36 achieve 47.8% and 48% respectively, and all settings outperform MGP-Short.

## 4.6 CONFIDENCE SCORE ANALYSIS

During inference, MGP uses confidence score-based masking (Fig. 2 and Fig. 3). To examine how this behaves in practice, we conducted two experiments. See Appendix Sec. C.5 for details.

First, we visualize token-wise confidence across refinement passes as actions are executed for MGP-Long in both the Meta-World and a dynamic environment. We show (i) a confidence heatmap and (ii) a matched mask–unmask map that marks which tokens are edited at each pass. On MetaWorld Disassemble (quasi-static, 'Very Hard'), confidence remains high during approach phases but becomes low during precise, outcome-critical manipulations and at actions requiring repeated attempts—for example, it stays high while the gripper approaches the ring, then falls when the gripper must position itself accurately to grasp and lift, especially when the first grasp fails and the policy makes repeated adjustments. On dynamic Basketball with a moving hoop, confidence is high while the hoop is stationary and drops once the hoop begins moving. Thus, drops in confidence typically align with environment changes, and refinement focuses on edits exactly where they matter.

Second, we assess calibration by resampling low- vs. high-confidence tokens on 'Disassemble' task in Metaworld. Masking the lowest 70% yields 0.86 success and a 60.6% flip rate (fraction of masked tokens that change after refinement), whereas masking the highest 70% reduces success and drops the flip rate to 15.1%, indicating that the scores reliably target uncertain tokens.

## 5 REAL WORLD EXPERIMENTS

To evaluate global planning in the real world, we deploy MGP and DP3 on a non-Markovian towel-sorting task. A towel and two baskets are placed in the robot's working area with a random lateral offset. At the beginning of each episode, a light briefly turns on in either red or blue and is then switched off; the color indicates which basket the towel should be placed in. The robot must grasp a towel and place it into the correct basket based solely on the initial light color. As the light is only visible at the very start of the episode, later observations do not contain any information about the desired goal, making the task non-Markovian. We collect 60 expert demonstrations by teleoperation of the LeRobot arm. For each trial, we record robot joint positions, end-effector position and orientation, gripper state, and synchronized RGB, depth, and point-cloud data. We evaluate our model in 25 real-world trials. Since short-horizon models such as DP3 and Quest are incapable of solving non-Markovian tasks (as shown in Sec. 4.4), we adopt DP3-FullSeq as the baseline. MGP-Long achieves a success rate of 96%, outperforming DP3-Full Seq (which achieves 84%). Moreover, MGP-Long maintains the same success rate when observations are missing during action execution. These results show that MGP-Long is robust to complex and noisy real-world conditions, and retains its strong global reasoning capabilities in this setting. See Appendix Sec. D for implementation details and qualitative results.

## 6 CONCLUSION

We have introduced Masked Generative Policy (MGP), the first masked-generative framework for visuomotor imitation learning, and two variations, MGP-Short for Markovian control and MGP-Long for complex and non-Markovian tasks. MGP-Short combines the sample diversity of diffusion methods with the low latency of autoregressive models, and delivers extremely rapid inference and improved success rates as evidenced in section 4. MGP-Long produces a full trajectory in one pass and then continuously refines future actions as new observations arrive; this enables global reasoning, dynamic adaptation, robustness to missing observations, and efficient and flexible replanning. Overall, our results position MGP as a fast, accurate, and flexible alternative to diffusion and visuomotor policies. However, MGP requires two-stage training, introducing potential mismatch and increased complexity. This limitation is left for future work.

## 7 REPRODUCIBILITY STATEMENT

Experiment details, including all hyperparameters and implementation details, are in Appendix Sec. A, with further results in Appendix Sec. C. These materials enable full reproducibility.

## ACKNOWLEDGEMENTS

This research has been supported by EPSRC Grant No. EP/S019472/1.

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

# Appendices

# A  EXPERIMENT DETAILS

## A.1  HYPERPARAMETERS

As for MGP-Short, Tab. 6 and Table 7 present the stage one and stage two of MGP-Short in Meta-World benchmark and in LIBERO benchmark.

As for MGP-Long, Tab. 8 presents stage one and stage two of MGP-Long in Meta-World benchmark, observation missing environments, dynamic environments and Non-Markovian environments.

Table 6: Stage 1 of MGP-Short in Meta-World and Libero Benchmark

| Parameter | Meta-World | Libero90&Libero-Long |
|---|---|---|
| gamma | 0.1 | 0.1 |
| commit | 0.02 | 0.02 |
| code dim | 16 | 16 |
| nb of code | 1024 | 8192 |
| down sampling rate | 2 | 2 |
| stride size | 2 | 2 |
| horizon | 8 | 32 |

Table 7: Stage 2 of MGP-Short in Meta-World and Libero Benchmark

| Parameter | Meta-World | Libero90&Libero-Long |
|---|---|---|
| gamma | 0.1 | 0.1 |
| weight decay | 1e-6 | 1e-6 |
| code dim | 16 | 16 |
| number of code | 1024 | 8192 |
| downsamping rate | 2 | 2 |
| stride size | 2 | 2 |
| block size | 54 | 68 |
| cross attention layer | 2 | 6 |
| self attention layer | 2 | 2 |
| embedding dimension | 256 | 512 |
| horizon | 8 | 32 |
| observation history | 4 | 1 |
| execution horizon | 5 | 8 |

## A.2  STANDARD BENCHMARKS

Meta-World (Yu et al., 2020) is a simulated benchmark with a broad set of robotic manipulation tasks. We use the official difficulty classification that ranks tasks from easy to very hard (Seo et al., 2023). Each demonstration contains 200 timesteps. In our experiments, we train 50 tasks on Meta-World separately.

We use LIBERO's language-conditioned manipulation suites in simulation (Liu et al., 2023). LIBERO-90 contains 90 single-goal, short-horizon tasks spanning diverse scenes (kitchen, living room, study) with rigid and articulated objects. Each task is accompanied by demonstrations (50 per task) with multi-modal observations (RGB views and proprioception). LIBERO-LONG is a long-duration benchmark which comprises 10 long-horizon compositions, each formed by sequencing two LIBERO-90 goals, requiring multi-stage execution.

Table 8: Stage 1 and Stage 2 of MGP-Long in Missing-obs, Dynamic and Non-Markovian Environments

| Stage one Parameter | Variant Environments | Stage two Parameter | Variant Environments |
|---|---|---|---|
| gamma | 0.1 | gamma | 0.1 |
| commit | 0.02 | weight decay | 1e-6 |
| code dim | 16 | code dim | 16 |
| nb of code | 1024 | number of code | 1024 |
| down sampling rate | 2 | downsamping rate | 2 |
| stride size | 2 | stride size | 2 |
| horizon | 128 | block size | 54 |
| | | cross attention layer | 2 |
| | | self attention layer | 2 |
| | | embedding dimension | 256 |
| | | observation history | 4 |
| | | execution horizon | 12 |

## A.3 IMPLEMENTATION DETAILS OF STANDARD BENCHMARKS

### A.3.1 SEPARATE TRAINING IN METAWORLD

As for the implementation details of MGP-short, we train a VQ-VAE tokenizer that maps every four consecutive primitive actions to a single discrete token in Stage 1. In Stage 2, a conditional masked transformer predicts two tokens (8 actions) in parallel, conditioning on observation features obtains from the same visual encoder as DP3 via cross-attention. At inference, MGP-Short conditions on the four most recent observations, predicts two tokens in parallel, performs two refinement iterations, and then executes the last five primitive actions before replanning.

As for implementation details of MGP-Long, the VQ-VAE tokenizer is first trained to map every 4 consecutive primitive actions to one discrete token over 128 actions. Afterwards, the decoded token sequences are truncated with different lengths. Among them, 70% retain the original length, while 30% are assigned a randomly sampled token length. The remaining training procedure is identical to that described in the main text. When inference, based on the initial four observations, MGP-Long predicts a full-horizon token sequence in a single forward pass (we use 50 tokens per episode). It then decodes and executes the next 12 primitive actions via the VQ decoder. After receiving the updated observations, the model recomputes confidence scores for the remaining tokens conditioned on the latest observations as the initial scores, then re-masks only the low-confidence subset (executed tokens and high-confidence tokens remain fixed), performs one selective refinement pass, and again decodes and executes the next 12 actions. This loop repeats until termination, yielding full-trajectory planning with efficient, on-the-fly adjustments while avoiding unnecessary regeneration of confident token segments.

### A.3.2 MULTITASK TRAINING IN LIBERO90

Training follows Sec A.3.1: A VQ-VAE tokenizer compresses 4 consecutive primitive actions to a single discrete token. It is followed by a conditional masked transformer condition on observation and task features via cross-attention, using the same encoders as QueST, and predicts 8 tokens (32 actions). At inference, MGP-Short conditions on the 1 most recent observations, predicts 4 tokens in parallel, performs two refinement iterations, and then executes the last 8 primitive actions.

## A.4 DYNAMIC ENVIRONMENTS SET UP

Five dynamic environments are designed based on Meta-World benchmark.

**Pick-Place-Wall (moving target)** In pick-place-wall environments, the goal site moves along $+x$ or $-x$ with equal probability by $0.14\,\mathrm{m}$ from steps 1–67 ($\sim 6.7\,\mathrm{s}$; $\approx 0.021\,\mathrm{m/s}$), then remains fixed.

**Basketball (moving net)** In basketball environment, the hoop body `basket_goal` moves along $+x$ by $0.10\,\mathrm{m}$ from steps 0–100 ($\sim 10\,\mathrm{s}$; $\approx 0.010\,\mathrm{m/s}$), then stays fixed.

**Pick-Place-Wall (moving wall)**  In pick-place-wall environment, the body `wall` moves continuously along $-x$ for the entire episode with total offset $0.20\,\text{m}$ (steps 0–200; $\approx 0.010\,\text{m/s}$)

**Push-Wall**  In push wall environment, the goal and `wall` translate together along $+x$ by $0.08\,\text{m}$ over steps 1–67 ($\sim 6.7\,\text{s}$; $\approx 0.012\,\text{m/s}$), then stop.

**Push**  In push environment, the goal marker (and object `objGeom`) translate along $+x$ by $0.10\,\text{m}$ during steps 1–67 ($\sim 6.7\,\text{s}$; $\approx 0.015\,\text{m/s}$), after which they remain stationary.

### A.5    Markovian environments

#### A.5.1    Task Description

There are two non-Markovian tasks: Button press on/off and Button press color change. In simulation, the workspace is a 80x80cm tabletop on top of which sits a modular rail that forms a closed square loop along the perimeter. The LeRobot SO101[2] is mounted on carriage that follows the rail. During an episode, the carriage moves clockwise or counter-clockwise to bring the arm into reach of different workspace.

(1) Button Press On/Off: Four push-buttons are placed 20cm away from each table corners; each has an LED ring with one of red, green, blue, yellow. At each episode start, colors are randomly reassigned to corners; all LEDs begin with the ON state. The robot must press buttons in the fixed color order red $\rightarrow$ green $\rightarrow$ blue $\rightarrow$ yellow, independent of location. A pressed button turns to the OFF state only while pressed and returns to the ON state on release. This causes the scene to look identical after a correct button press, rendering progress unobservable from a single frame (non-Markovian). Success is defined by pressing all four colors in order within the horizon.

(2) Button Press Color Change. Same layout as the previous task, but each button cycles through five states on every press: yellow $\rightarrow$ red $\rightarrow$ green $\rightarrow$ blue $\rightarrow$ off. The four buttons start with distinct initial colors. The task is to press buttons in the order of their initial colors (red, then green, then blue, then yellow). Thus multiple buttons may display the same color during execution (e.g. after pressing the green button, there will be two blue buttons), and identical frames can correspond to different stages, again non-Markovian. Success is recorded when the all the buttons have been pressed within the horizon.

#### A.5.2    Dataset

We collect 150 demonstrations per task. Each demonstration contains synchronized RGB images, depth maps, and colorized point clouds, along with robot proprioception and end-effector (EE) signals. For each demonstration, we record approximately 400-500 timesteps. Point clouds are first cropped to the workspace and then downsampled to 1,024 points. Robot proprioception includes joint positions of robot, base position relative to track and base rotation relative to track. EE information includes EE position, EE orientation, and gripper open/closed state. We include videos of several demonstrations in the supplementary files.

#### A.5.3    Implementation Details

Inputs are categorized into (1) observations and (2) robot state. For DP3 baseline, we use downsampled point cloud as observation and for Quest, we use RGB images. Robot state, used as an additional conditioning input, include joint states, base position and rotation relative to track, EE position, EE orientation, and gripper open/closed state. The model outputs an action at each timestep consisting of the target EE position, EE orientation, and a gripper command.

## B    Ablation Studies and Hyperparameter Experiments

We conducted seven ablation studies, three of which focused on MGP-Short and four on MGP-Long. For MGP-Short, the studies included: (1) the refinement step size; (2) the tokenizer's codebook size; and (3) the discretization granularity. For MGP-Long, the studies included: (1) the refinement step

---

[2]https://github.com/TheRobotStudio/SO-ARM100

Table 9: Success rates of MGP-Short under different refinement steps across Meta-World Hard tasks.

| | **Hard (5)** | | | | | |
|---|---|---|---|---|---|---|
| **steps** | Assembly | Hand insert | Pick out of hole | Pick place | Push | **Average** |
| step=1 | **1.00** | 0.15 | 0.25 | 0.20 | 0.38 | 0.396 |
| step=2 | **1.00** | **0.29** | **0.58** | **0.30** | **0.53** | **0.540** |

Table 10: Success rates of MGP-Short under different tokenizer's codebook size across Meta-World Very Hard tasks.

| | **Very Hard (5)** | | | | | |
|---|---|---|---|---|---|---|
| **Codebook Size** | Shelf place | Disassemble | Stick pull | Stick push | Pick place wall | **Average** |
| 512 | **0.23** | **0.81** | 0.41 | 0.87 | 0.35 | 0.534 |
| 1024 | 0.20 | 0.74 | **0.50** | **0.90** | 0.35 | **0.538** |
| 2048 | 0.21 | 0.80 | 0.39 | 0.85 | **0.36** | 0.522 |

size under challenging conditions; (2) the mask ratio; (3) the scoring strategy; and (4) different execution step sizes.

## B.1 REFINE STEP FOR MGP-SHORT

We ablate the number of mask–refine iterations $r \in \{1, 2, 3\}$ for MGP-Short on five Meta-World Hard tasks. As shown in Tab. 9, increasing from $r = 1$ to $r = 2$ yields a success gain of 14.3%, proving the effectiveness of *score-based masking scheme*, whereas $r = 3$ provides no statistically significant additional improvement, while incurring higher inference cost. We therefore adopt $r = 2$ as the default for short-timestep tasks, striking a favorable accuracy–latency trade-off.

## B.2 CODEBOOK SIZE FOR MGP-SHORT

We explore the influence of codebook size on five Meta-World Very Hard tasks, training separate models with codebook sizes of 512, 1024, and 2048. The average success rates are 0.534, 0.538, and 0.522, respectively. The gaps are small (1.6%) and show no consistent trend, indicating minimal sensitivity to codebook size. Detailed results are shown in Tab. 10.

## B.3 DISCRETIZATION GRANULARITY FOR MGP-SHORT

We test with 2 actions/token and 8 actions/token on the five Meta-World Very Hard tasks. Averaged over tasks, 4 actions/token achieves the highest success rate (0.538), outperforming 2 actions/token (0.526, +1.2%) and 8 actions/token (0.514, +2.4%). The gaps are modest, indicating limited sensitivity to granularity. Detailed results are shown in Tab. 11.

## B.4 REFINE STEP FOR MGP-LONG UNDER CHALLENGING CONDITIONS

We evaluated $r \in \{1, 2, 3\}$ on five dynamic tasks of MGP-Long. Increasing the refinement steps from $r = 1$ to $r = 2$ raises the success rate by +5.2%, indicating that the score-based masking scheme is helpful in more challenging settings. Increasing further from $r = 2$ to $r = 3$ yields $< 1\%$ improvement, while incurring higher inference cost. We therefore set $r = 2$ for MGP-Long in the main experiments. Detailed results are shown in Tab. 12.

## B.5 MASK RATIO FOR MGP-LONG

We vary what proportion of low confidence tokens are resampled. When replanning we rank the unexecuted tokens by posterior confidence and in our standard setting mask the bottom 70% for one refinement pass. Changing between 50%, 70%, and 85% shows that 70% yields the best average success across five Meta-World Very Hard tasks. A 50% ratio underperforms because many low-

Table 11: Success rates of MGP-Short under different Discretization granularity across Meta-World Very Hard tasks.

| Method | Very Hard (5) | | | | | |
| | Shelf place | Disassemble | Stick pull | Stick push | Pick place wall | Average |
|---|---|---|---|---|---|---|
| 2 actions/token | 0.21 | **0.90** | 0.33 | 0.88 | 0.31 | 0.526 |
| 4 actions/token | 0.20 | 0.74 | **0.50** | **0.90** | **0.35** | **0.538** |
| 8 actions/token | **0.25** | 0.78 | 0.35 | 0.86 | 0.33 | 0.514 |

Table 12: Success rates of MGP-Long under different refine step across dynamic environments.

| Method | Dynamic Environment (5) | | | | | |
| | Basketball | Pick place wall(W) | Pick place wall(T) | Push | Push wall(T) | Average |
|---|---|---|---|---|---|---|
| 1 | **1.00** | **0.31** | 0.08 | 0.15 | 0.38 | 0.384 |
| 2 | **1.00** | 0.30 | **0.13** | **0.25** | **0.50** | **0.436** |
| 3 | **1.00** | **0.31** | 0.10 | 0.20 | 0.48 | 0.418 |

confidence tokens remain unedited and can also interfere with accurate regeneration of the masked ones; 85% is comparable to 70% on average. Detailed results are shown in Tab. 13.

### B.6 SCORE POLICY FOR MGP-LONG

Token selection for re-masking is critical in MGP-Long as it determines how effectively high-confidence tokens are reused and focuses computation on truly uncertain positions for targeted refinement. We ablate three confidence–update schemes, applied at each replanning step $t$, to choose which future tokens to re-mask first *prior to refinement* : (1) *Random*: Use random scores to mask the rest of the tokens. (2) *Score Reuse*: Reuse the previous confidences $s_{t-1}$ from last steps to select the lowest-confidence tokens. (3) *ATR*: Recompute current confidences $s_t$ from the latest observations using the masked generative transformer (via cross attention) first, then select only the presently low-confidence tokens. We evaluate on ten Meta-World Hard and Very Hard tasks, reporting the average success rate. As shown in Tab. 14, *ATR* scheme attains the highest success, increasing by 10.68% over *Random* scores and improving by 5.53% from *Score Reuse*. *Score reuse* scheme underperforms because its confidences become stale and miscalibrated after environment changes, whereas random score performs worst because it ignores uncertainty and observation cues, re-masking tokens arbitrarily. We therefore adopt *ATR* scheme before refinement as the default scoring policy for MGP-Long.

### B.7 VARYING EXECUTION STEP

On five hard Meta-World tasks, we evaluate MGP-Long across execution step sizes of 4, 12, and 36. A step size of 12 yields the highest success (54%), while steps of 4 and 36 achieve 47.8% and 48% respectively, and all settings outperform MGP-Short. Detailed results are shown in Tab. 15

## C ADDITIONAL SIMULATION RESULTS

### C.1 EVALUATION ON STANDARD BENCHMARKS

#### C.1.1 SINGLE TASK TRAINING OF METAWORLD

We evaluate MGP-Short on 50 tasks in Meta-World benchmark across different difficult levels. Detailed results are shown in Tab. 16

We further evaluate MGP-Long on ten Hard and Very Hard tasks. Detailed results are shown in Tab. 17.

Table 13: Success rates of MGP-Long under different mask ratios across Meta-World Very Hard tasks.

| | Very Hard (5) | | | | | |
|---|---|---|---|---|---|---|
| **Mask Ratio** | Shelf place | Disassemble | Stick pull | Stick push | Pick place wall | **Average** |
| 50% | 0.25 | 0.83 | 0.41 | **1.00** | 0.31 | 0.560 |
| 70% | **0.29** | **0.86** | **0.45** | **1.00** | **0.33** | **0.586** |
| 85% | 0.27 | 0.85 | 0.45 | **1.00** | 0.31 | 0.576 |

Table 14: Success rates of MGP-Long under different scoring policies across Meta-World Hard and Very Hard tasks.

| | Hard (5) | | | | | |
|---|---|---|---|---|---|---|
| **Method** | Assembly | Hand insert | Pick out of hole | Pick place | Push | **Average** |
| MGP-Long w.rs | **1.00** | 0.16 | 0.42 | 0.12 | 0.47 | 0.434 |
| MGP-Long w.ls | **1.00** | 0.22 | 0.47 | 0.15 | 0.50 | 0.468 |
| **MGP-Long w.ns (ours)** | **1.00** | **0.29** | **0.58** | **0.30** | **0.53** | **0.540** |
| | Very Hard (5) | | | | | |
| **Method** | Shelf place | Disassemble | Stick pull | Stick Push | Pick place Wall | **Average** |
| MGP-Long w.rs | 0.11 | 0.75 | 0.32 | 0.95 | 0.26 | 0.478 |
| MGP-Long w.ls | 0.25 | 0.76 | 0.40 | **1.00** | 0.32 | 0.546 |
| **MGP-Long w.ns (ours)** | **0.29** | **0.86** | **0.45** | **1.00** | **0.33** | **0.586** |

### C.1.2 MULTITASK TRAINING OF LIBERO90

We evaluate MGP-Short across all 90 tasks of LIBERO-90 in multi-task training setting. Detailed results are shown in Tab. 18

### C.1.3 MULTITASK TRAINING OF LIBERO10

In the LIBERO-10 benchmark, the task IDs correspond as follows:

Task 1 (Study_SCENE1)) pick up the book and place it in the back compartment of the caddy;

Task 2 (LIVING_ROOM_SCENE6) – put the white mug on the plate and then place the chocolate pudding to the right of the plate;

Task 3 (LIVING_ROOM_SCENE5) – put the white mug on the left plate and the yellow-and-white mug on the right plate;

Task 4 (LIVING_ROOM_SCENE2) – put both the cream cheese box and the butter in the basket;

Task 5 (LIVING_ROOM_SCENE2) – put both the alphabet soup and the tomato sauce in the basket;

Task 6 (LIVING_ROOM_SCENE1) – place both the alphabet soup and the cream cheese box in the basket;

Task 7 (KITCHEN_SCENE8) – put both moka pots on the stove;

Task 8 (KITCHEN_SCENE6) – put the yellow-and-white mug in the microwave and close it;

Task 9 (KITCHEN_SCENE4) – put the black bowl in the bottom drawer of the cabinet and close it;

Task 10 (KITCHEN_SCENE3) – turn on the stove and put the moka pot on it.

Detailed results are shown in Tab. 19.

Table 19: Success Rate of Multitask training on Libero-10.

| Task ID | 1 | 2 | 3 | 4 | 5 | 6 | 7 | 8 | 9 | 10 | Average SR |
|---|---|---|---|---|---|---|---|---|---|---|---|
| **MGP-Short (ours)** | 0.78 | 0.66 | 0.60 | 0.86 | 0.48 | 0.92 | 0.78 | 0.78 | 0.92 | 0.92 | 0.770 |
| **MGP-Long (ours)** | 0.81 | 0.70 | 0.73 | 0.90 | 0.55 | 0.94 | 0.86 | 0.83 | 0.95 | 0.93 | 0.820 |

Table 15: Success rates of MGP-Long under different execution steps across Meta-World Hard tasks.

| Method | Assembly | Hand insert | Pick out of hole | Pick place | Push | Average |
|---|---|---|---|---|---|---|
| | **Hard (5)** | | | | | |
| step=4 | **1.00** | 0.27 | 0.50 | 0.25 | 0.37 | 0.478 |
| step=12 | **1.00** | **0.29** | **0.58** | **0.30** | **0.53** | **0.540** |
| step=36 | **1.00** | 0.24 | 0.48 | 0.25 | 0.43 | 0.480 |

Table 16: Success rates of Diffusion Policy, 3D Diffusion Policy and MGP-Short (ours) across all tasks of Meta-World.

| Alg&Task | Button Press | Button Press Topdown | Button Press Topdown Wall | Button Press Wall | Coffee Button | Dial Turn |
|---|---|---|---|---|---|---|
| | **Easy (28)** | | | | | |
| DP | 0.99 | 0.98 | 0.96 | 0.97 | 0.99 | 0.63 |
| DP3 | **1.00** | **1.00** | 0.99 | 0.99 | **1.00** | **0.66** |
| MGP-Short (ours) | **1.00** | **1.00** | **1.00** | **1.00** | **1.00** | 0.65 |

| Alg&Task | Door Close | Door Lock | Door open | Door unlock | Drawer close | Drawer open |
|---|---|---|---|---|---|---|
| | **Easy (28)** | | | | | |
| DP | **1.00** | 0.86 | 0.98 | 0.98 | **1.00** | 0.93 |
| DP3 | **1.00** | 0.98 | 0.99 | **1.00** | **1.00** | **1.00** |
| MGP-Short (ours) | **1.00** | **1.00** | **1.00** | **1.00** | **1.00** | **1.00** |

| Alg&Task | Faucet close | Faucet open | Handle press | Handle pull | Handle press side | Handle pull side |
|---|---|---|---|---|---|---|
| | **Easy (28)** | | | | | |
| DP | **1.00** | **1.00** | 0.81 | 0.27 | **1.00** | 0.23 |
| DP3 | **1.00** | **1.00** | **1.00** | 0.53 | **1.00** | **0.85** |
| MGP-Short (ours) | **1.00** | **1.00** | **1.00** | **0.65** | **1.00** | **0.85** |

| Alg&Task | lever pull | plate slide | plate slide back | plate slide back side | plate slide side | Reach wall |
|---|---|---|---|---|---|---|
| | **Easy (28)** | | | | | |
| DP | 0.49 | 0.83 | 0.99 | **1.00** | **1.00** | 0.59 |
| DP3 | **0.79** | **1.00** | 0.99 | **1.00** | **1.00** | 0.68 |
| MGP-Short (ours) | 0.62 | **1.00** | **1.00** | **1.00** | **1.00** | **0.75** |

| Alg&Task | window close | window open | peg unplug side | reach | Average | |
|---|---|---|---|---|---|---|
| | **Easy (28)** | | | | | |
| DP | **1.00** | **1.00** | 0.74 | 0.18 | 0.836 | |
| DP3 | **1.00** | **1.00** | 0.75 | 0.24 | 0.909 | |
| MGP-Short (ours) | **1.00** | **1.00** | **0.80** | **0.44** | **0.920** | |

| Alg&Task | Basketball | Bin picking | Box close | Coffee pull | Coffee Push | Hammer |
|---|---|---|---|---|---|---|
| | **Medium (11)** | | | | | |
| DP | 0.85 | 0.15 | 0.30 | 0.34 | 0.67 | 0.15 |
| DP3 | 0.98 | **0.34** | 0.42 | **0.87** | **0.94** | 0.76 |
| MGP-Short (ours) | **1.00** | 0.24 | **0.57** | **0.87** | 0.87 | **0.89** |

| Alg&Task | Peg insert side | Push wall | soccer | sweep | sweep into | Average |
|---|---|---|---|---|---|---|
| | **Medium (11)** | | | | | |
| DP | 0.34 | 0.20 | 0.14 | 0.18 | 0.10 | 0.311 |
| DP3 | **0.69** | 0.49 | 0.18 | **0.96** | 0.15 | 0.616 |
| MGP-Short (ours) | 0.49 | **0.77** | **0.40** | 0.70 | **0.39** | **0.650** |

| Alg&Task | Assembly | Hand insert | Pick out of hole | Pick place | Push | Average |
|---|---|---|---|---|---|---|
| | **Hard (5)** | | | | | |
| DP | 0.15 | 0.09 | 0.00 | 0.00 | 0.30 | 0.108 |
| DP3 | 0.99 | 0.14 | 0.14 | 0.12 | 0.51 | 0.380 |
| MGP (ours) | **1.00** | **0.19** | **0.15** | **0.35** | **0.52** | **0.440** |

| Alg&Task | Shelf place | Disassemble | Stick pull | Stick Push | Pick place Wall | Average |
|---|---|---|---|---|---|---|
| | **Very Hard (5)** | | | | | |
| DP | 0.11 | 0.43 | 0.11 | 0.63 | 0.05 | 0.266 |
| DP3 | 0.17 | 0.69 | 0.27 | **0.97** | **0.35** | 0.490 |
| MGP-Short (ours) | **0.2** | **0.74** | **0.50** | 0.90 | **0.35** | **0.538** |

## C.2 EVALUATION ON OBSERVATION-MISSING ENVIRONMENTS

We evaluate MGP-Short and MGP-Long in observation missing environments. Detailed results are shown in Tab. 20.

From the results, we can see as the missing-observation rate increases, performance degrades gracefully. With p=0.35/0.50/0.70 (where p is the probability that missing observations occur for each rollout), MGP-Long achieves 0.484/0.466/0.462 success rate on Hard and 0.564/0.550/0.566 success rate on Very Hard. Overall, the performance remains stable even when observations are missing

Table 17: Success rates of 3D Diffusion Policy, MGP-Short (ours), Full sequence DP3, Full sequence MGP, MGP without scored-based mask and MGP-Long (ours) across Hard and Very Hard tasks of Meta-World.

| Alg&Task | Hard (5) | | | | | |
|---|---|---|---|---|---|---|
| | Assembly | Hand insert | Pick out of hole | Pick place | Push | **Average** |
| DP3 | 0.99 | 0.14 | 0.14 | 0.12 | 0.51 | 0.380 |
| **MGP-Short (ours)** | 1.00 | 0.19 | 0.15 | **0.35** | 0.52 | 0.440 |
| DP3-Full Seq. | 0.20 | 0.17 | 0.25 | 0.12 | 0.20 | 0.188 |
| MGP-Full Seq. | 0.30 | 0.18 | 0.50 | 0.14 | 0.35 | 0.294 |
| MGP-w/o SM | **1.00** | 0.27 | 0.49 | 0.30 | 0.49 | 0.510 |
| **MGP-Long (ours)** | **1.00** | **0.29** | **0.58** | 0.30 | **0.53** | **0.540** |
| Alg&Task | Very Hard (5) | | | | | |
| | Shelf place | Disassemble | Stick pull | Stick Push | Pick place Wall | **Average** |
| DP3 | 0.17 | 0.69 | 0.27 | 0.97 | **0.35** | 0.490 |
| **MGP-Short (ours)** | 0.20 | 0.74 | **0.50** | 0.90 | **0.35** | 0.538 |
| DP3-Full Seq. | 0.15 | 0.63 | 0.10 | 0.67 | 0.20 | 0.350 |
| MGP-Full Seq. | 0.19 | 0.75 | 0.12 | 0.72 | 0.15 | 0.386 |
| MGP-w/o SM | 0.28 | 0.83 | 0.44 | **1.00** | 0.31 | 0.572 |
| **MGP-Long (ours)** | **0.29** | **0.86** | 0.45 | **1.00** | 0.33 | **0.586** |

most of the time and consistently surpasses short-horizon baselines. Even at p=0.70, MGP-Long exceeds MGP-Short by 30.9% and DP3 by 31.5%, demonstrating strong robustness to high missing-observation rates.

## C.3 EVALUATION ON DYNAMIC ENVIRONMENTS

Qualitative results of MGP-Long are shown in Fig. 4. The figure covers five dynamic scenarios:(a) a moving target in *Push (Wall)*; (b) a moving goal location in *Pick-Place (Wall)*; (c) a moving target in *Push*; (d) a moving hoop/net in the *Basketball* environment and (e) a moving wall (obstacle) in *Pick-Place (Wall)*. We provide videos visualizing the results of all dynamic environments in the supplementary material.

**Analysis of failure cases.** In one-shot methods (DP3-Full / MGP-Full), the plan is fixed at $t = 0$, i.e. there are no closed-loop updates; thus, when the world changes, the planned action trajectory becomes stale, leading to missed contacts, wrong targets, and compounding errors. Short-horizon DP3, with its tiny context window, is overly reactive: it lags moving objects, overfits local noise, and can oscillate when the task needs longer-range context or memory. For MGP-Long (ours), most failures arise when changes outpace the replan cadence or the required edits exceed the current mask window; perception latency or occlusion can also delay corrections. For example, in dynamic *Push* with a moving goal, the robot briefly chases the old position and zigzags until confidence drops and that segment is rewritten.

## C.4 EVALUATION ON NON-MARKOVIAN ENVIRONMENTS

Qualitative results of MGP-Long and short-horizon baselines of Button Press On/Off and Button Press Change Color task are shown in Fig. 5 and Fig. 6, respectively. From the qualitative results, we can see that MGP-Long can press the buttons with different colors in the right order. However, short-horizon baselines often stall or press the wrong buttons. We provide videos visualizing the results of MGP-Long and short-horizon baselines across all Non-Markovian environments in the supplementary material.

**Analysis of failure cases.** The failures come from non-Markovian aliasing and myopic control: once the brief color cue disappears, subsequent observations no longer reveal the latent goal, and policies that rely on a short temporal context window (without long timestep memory or explicit plan) either hesitate near the panel or guess a plausible—but incorrect—sequence. A single mis-press then changes the hidden state; without a mechanism to replan, the controller falls into press–unpress oscillations or repeats the same action.

Table 18: Success Rate and Inference Time of Multitask training on Libero-90. Unit of inference time is ms/step.

| Task ID | Success Rate | | | | | | |
|---|---|---|---|---|---|---|---|
| | **ResNet-T** | **ACT** | **PRISE** | **VQ-BeT** | **Diffusion Policy** | **Quest** | **MGP-Short (ours)** |
| 1 | 1.00 | 0.90 | 0.80 | 1.00 | 0.99 | 1.00 | 0.97 |
| 2 | 0.96 | 0.30 | 0.35 | 0.94 | 0.98 | 0.97 | 0.97 |
| 3 | 0.96 | 0.50 | 0.70 | 0.97 | 0.99 | 0.91 | 0.97 |
| 4 | 0.74 | 0.22 | 0.50 | 0.99 | 0.91 | 0.94 | 0.77 |
| 5 | 0.95 | 0.58 | 0.45 | 0.95 | 0.93 | 0.98 | 0.97 |
| 6 | 0.91 | 0.39 | 0.65 | 0.98 | 0.99 | 0.98 | 0.97 |
| 7 | 0.95 | 0.29 | 0.50 | 0.86 | 0.94 | 0.93 | 0.87 |
| 8 | 0.96 | 0.72 | 0.95 | 0.80 | 0.90 | 0.99 | 0.90 |
| 9 | 0.74 | 0.41 | 0.93 | 0.60 | 0.75 | 0.93 | 0.87 |
| 10 | 0.97 | 0.65 | 0.35 | 0.83 | 0.91 | 0.90 | 1.00 |
| 11 | 0.97 | 0.82 | 0.95 | 0.96 | 0.98 | 0.97 | 1.00 |
| 12 | 0.90 | 0.73 | 0.95 | 0.80 | 0.94 | 0.94 | 0.93 |
| 13 | 0.82 | 0.62 | 0.20 | 0.87 | 0.81 | 0.76 | 0.87 |
| 14 | 0.86 | 0.72 | 0.40 | 0.49 | 0.94 | 0.71 | 0.80 |
| 15 | 0.87 | 0.49 | 0.35 | 0.46 | 0.95 | 0.58 | 0.64 |
| 16 | 0.97 | 0.86 | 0.75 | 0.98 | 0.99 | 0.96 | 0.93 |
| 17 | 0.72 | 0.40 | 0.40 | 0.53 | 0.89 | 0.80 | 0.77 |
| 18 | 0.79 | 0.20 | 0.15 | 0.80 | 0.76 | 0.67 | 0.67 |
| 19 | 0.83 | 0.75 | 0.30 | 0.91 | 0.99 | 1.00 | 0.90 |
| 20 | 0.87 | 0.41 | 0.65 | 0.68 | 0.99 | 1.00 | 0.90 |
| 21 | 1.00 | 0.82 | 1.00 | 0.96 | 0.99 | 1.00 | 1.00 |
| 22 | 0.99 | 0.44 | 0.30 | 0.91 | 0.97 | 0.93 | 0.97 |
| 23 | 0.97 | 0.75 | 0.85 | 0.95 | 0.99 | 0.92 | 0.97 |
| 24 | 0.75 | 0.90 | 0.80 | 1.00 | 0.99 | 1.00 | 0.90 |
| 25 | 0.97 | 0.44 | 0.95 | 0.94 | 0.99 | 1.00 | 1.00 |
| 26 | 0.97 | 0.85 | 0.90 | 0.85 | 1.00 | 0.99 | 0.97 |
| 27 | 0.72 | 0.14 | 0.55 | 0.50 | 0.88 | 0.52 | 0.70 |
| 28 | 0.72 | 0.20 | 0.05 | 0.45 | 0.86 | 0.68 | 0.60 |
| 29 | 1.00 | 0.68 | 1.00 | 0.97 | 1.00 | 1.00 | 1.00 |
| 30 | 1.00 | 0.19 | 1.00 | 0.92 | 1.00 | 0.97 | 1.00 |
| 31 | 0.91 | 0.83 | 0.50 | 0.85 | 0.96 | 0.90 | 0.97 |
| 32 | 0.99 | 0.90 | 0.85 | 0.88 | 1.00 | 0.99 | 1.00 |
| 33 | 0.57 | 0.20 | 0.20 | 0.37 | 0.58 | 0.67 | 0.67 |
| 34 | 0.85 | 0.56 | 0.30 | 0.87 | 0.84 | 0.98 | 0.97 |
| 35 | 0.93 | 0.52 | 0.80 | 0.98 | 0.97 | 0.92 | 0.90 |
| 36 | 0.97 | 0.67 | 0.75 | 0.98 | 0.99 | 0.97 | 0.97 |
| 37 | 0.85 | 0.24 | 0.25 | 0.73 | 0.97 | 0.74 | 0.80 |
| 38 | 0.78 | 0.41 | 0.30 | 0.90 | 0.91 | 0.62 | 0.67 |
| 39 | 0.86 | 0.32 | 0.20 | 0.90 | 0.90 | 0.88 | 0.90 |
| 40 | 0.96 | 0.35 | 0.85 | 0.90 | 0.98 | 0.93 | 0.93 |
| 41 | 0.90 | 0.27 | 0.50 | 0.91 | 0.79 | 0.92 | 0.93 |
| 42 | 1.00 | 0.74 | 0.55 | 0.89 | 1.00 | 1.00 | 1.00 |
| 43 | 0.98 | 0.41 | 0.80 | 0.97 | 0.99 | 0.98 | 0.97 |
| 44 | 0.80 | 0.39 | 0.40 | 0.83 | 0.89 | 0.93 | 0.87 |
| 45 | 0.99 | 0.83 | 0.85 | 0.99 | 1.00 | 0.98 | 0.93 |

| Task ID | Success Rate | | | | | | |
| --- | --- | --- | --- | --- | --- | --- | --- |
| | ResNet-T | ACT | PRISE | VQ-BeT | Diffusion Policy | Quest | MGP (ours) |
| 46 | 0.97 | 0.60 | 0.55 | 0.91 | 1.00 | 0.99 | 1.00 |
| 47 | 0.75 | 0.37 | 0.35 | 0.65 | 0.31 | 0.91 | 0.83 |
| 48 | 0.87 | 0.27 | 0.25 | 0.88 | 0.53 | 0.98 | 1.00 |
| 49 | 0.90 | 0.55 | 0.65 | 0.48 | 0.96 | 0.95 | 0.90 |
| 50 | 0.88 | 0.54 | 0.65 | 0.60 | 0.82 | 0.99 | 0.78 |
| 51 | 0.80 | 0.33 | 0.40 | 0.87 | 0.28 | 0.88 | 0.70 |
| 52 | 0.79 | 0.28 | 0.10 | 0.91 | 0.00 | 0.75 | 0.70 |
| 53 | 0.74 | 0.33 | 0.30 | 0.84 | 0.34 | 0.82 | 0.83 |
| 54 | 0.88 | 0.64 | 0.60 | 0.79 | 0.73 | 0.87 | 0.83 |
| 55 | 0.83 | 0.51 | 0.50 | 0.95 | 0.77 | 0.93 | 0.97 |
| 56 | 0.85 | 0.62 | 0.35 | 0.92 | 0.49 | 0.83 | 0.93 |
| 57 | 0.99 | 0.64 | 0.80 | 1.00 | 1.00 | 0.97 | 0.93 |
| 58 | 0.95 | 0.57 | 0.50 | 1.00 | 1.00 | 0.99 | 1.00 |
| 59 | 0.84 | 0.56 | 0.20 | 0.98 | 0.78 | 0.95 | 0.87 |
| 60 | 0.94 | 0.68 | 0.65 | 0.91 | 0.89 | 1.00 | 0.97 |
| 61 | 0.91 | 0.95 | 0.80 | 0.98 | 0.90 | 1.00 | 1.00 |
| 62 | 0.96 | 0.75 | 0.85 | 0.99 | 0.58 | 0.81 | 0.97 |
| 63 | 0.70 | 0.43 | 0.40 | 0.84 | 0.38 | 0.78 | 0.90 |
| 64 | 0.73 | 0.04 | 0.40 | 0.38 | 0.41 | 0.78 | 0.70 |
| 65 | 0.73 | 0.16 | 0.15 | 0.68 | 0.75 | 0.85 | 0.90 |
| 66 | 0.76 | 0.45 | 0.15 | 0.84 | 0.65 | 0.79 | 0.77 |
| 67 | 0.84 | 0.72 | 0.30 | 0.87 | 0.66 | 0.93 | 0.90 |
| 68 | 0.78 | 0.73 | 0.55 | 0.74 | 0.44 | 0.85 | 0.93 |
| 69 | 0.83 | 0.68 | 0.85 | 0.90 | 0.59 | 0.93 | 0.90 |
| 70 | 0.88 | 0.56 | 0.90 | 0.93 | 0.57 | 0.89 | 0.93 |
| 71 | 0.90 | 0.52 | 0.55 | 0.97 | 0.92 | 0.92 | 0.97 |
| 72 | 0.85 | 0.52 | 0.35 | 0.85 | 0.98 | 0.94 | 0.87 |
| 73 | 0.89 | 0.59 | 0.60 | 0.84 | 0.86 | 0.98 | 0.77 |
| 74 | 0.72 | 0.18 | 0.30 | 0.33 | 0.61 | 0.70 | 0.77 |
| 75 | 0.77 | 0.45 | 0.45 | 0.95 | 0.38 | 0.95 | 1.00 |
| 76 | 0.64 | 0.22 | 0.25 | 0.30 | 0.21 | 0.61 | 0.73 |
| 77 | 0.89 | 0.70 | 0.65 | 0.70 | 0.35 | 0.89 | 0.87 |
| 78 | 0.57 | 0.46 | 0.80 | 0.85 | 0.14 | 0.97 | 0.97 |
| 79 | 0.63 | 0.28 | 0.45 | 0.68 | 0.06 | 0.86 | 0.84 |
| 80 | 0.73 | 0.59 | 0.30 | 0.87 | 0.01 | 0.98 | 1.00 |
| 81 | 0.65 | 0.53 | 0.30 | 0.44 | 0.08 | 0.70 | 0.80 |
| 82 | 0.63 | 0.24 | 0.35 | 0.61 | 0.54 | 0.70 | 0.73 |
| 83 | 0.80 | 0.56 | 0.80 | 0.89 | 0.49 | 0.94 | 0.97 |
| 84 | 0.55 | 0.35 | 0.55 | 0.43 | 0.47 | 0.75 | 0.87 |
| 85 | 0.70 | 0.74 | 0.75 | 0.93 | 0.79 | 0.92 | 1.00 |
| 86 | 0.69 | 0.53 | 0.75 | 0.47 | 0.13 | 0.89 | 1.00 |
| 87 | 0.84 | 0.65 | 0.95 | 0.86 | 0.98 | 0.92 | 0.87 |
| 88 | 0.82 | 0.54 | 0.65 | 0.87 | 0.96 | 0.97 | 1.00 |
| 89 | 0.91 | 0.77 | 0.55 | 0.96 | 0.70 | 0.97 | 0.90 |
| 90 | 0.80 | 0.29 | 0.85 | 0.89 | 0.91 | 0.56 | 0.70 |

Table 20: Success rates of 3D Diffusion Policy, MGP-Short (ours), Full sequence DP3, Full sequence MGP, MGP without scored-based mask and MGP-Long (ours) across Hard and Very Hard tasks of Meta-World under Observation-missing environments.

| Alg&Task | Hard (5) | | | | | |
| --- | --- | --- | --- | --- | --- | --- |
| | Assembly | Hand insert | Pick out of hole | Pick place | Push | **Average** |
| DP3 (p=0.35) | 0.15 | 0.16 | 0.05 | **0.20** | 0.24 | 0.160 |
| **MGP-Short (ours)** (p=0.35) | 0.15 | 0.20 | 0.05 | 0.17 | 0.29 | 0.172 |
| DP3-Full Seq. | 0.20 | 0.17 | 0.25 | 0.12 | 0.20 | 0.188 |
| MGP-Full Seq. | 0.30 | 0.18 | 0.50 | 0.14 | 0.35 | 0.294 |
| MGP-w/o SM (p=0.35) | 0.97 | 0.21 | 0.48 | 0.17 | 0.25 | 0.416 |
| **MGP-Long (ours)(p=0.35)** | **1.00** | **0.23** | 0.54 | **0.20** | **0.45** | **0.484** |
| MGP-Long (ours)(p=0.50) | **1.00** | 0.20 | 0.53 | **0.20** | 0.40 | 0.466 |
| MGP-Long (ours)(p=0.70) | **1.00** | 0.18 | **0.57** | 0.18 | 0.38 | 0.462 |

| Alg&Task | Very Hard (5) | | | | | |
| --- | --- | --- | --- | --- | --- | --- |
| | Shelf place | Disassemble | Stick pull | Stick Push | Pick place Wall | **Average** |
| DP3 | 0.12 | 0.30 | 0.18 | 0.37 | 0.23 | 0.240 |
| **MGP-Short (ours)** | 0.15 | 0.27 | 0.16 | 0.41 | 0.20 | 0.238 |
| DP3-Full Seq. | 0.15 | 0.63 | 0.10 | 0.67 | 0.20 | 0.350 |
| MGP-Full Seq. | 0.19 | 0.75 | 0.12 | 0.72 | 0.15 | 0.386 |
| MGP-w/o SM(p=0.35) | 0.20 | 0.84 | 0.39 | 0.96 | 0.30 | 0.538 |
| **MGP-Long (ours)(p=0.35)** | 0.20 | **0.87** | **0.43** | **1.00** | **0.32** | 0.564 |
| MGP-Long (ours)(p=0.50) | 0.22 | 0.82 | 0.41 | **1.00** | 0.30 | 0.550 |
| MGP-Long (ours)(p=0.70) | **0.24** | 0.86 | 0.42 | **1.00** | 0.31 | **0.566** |

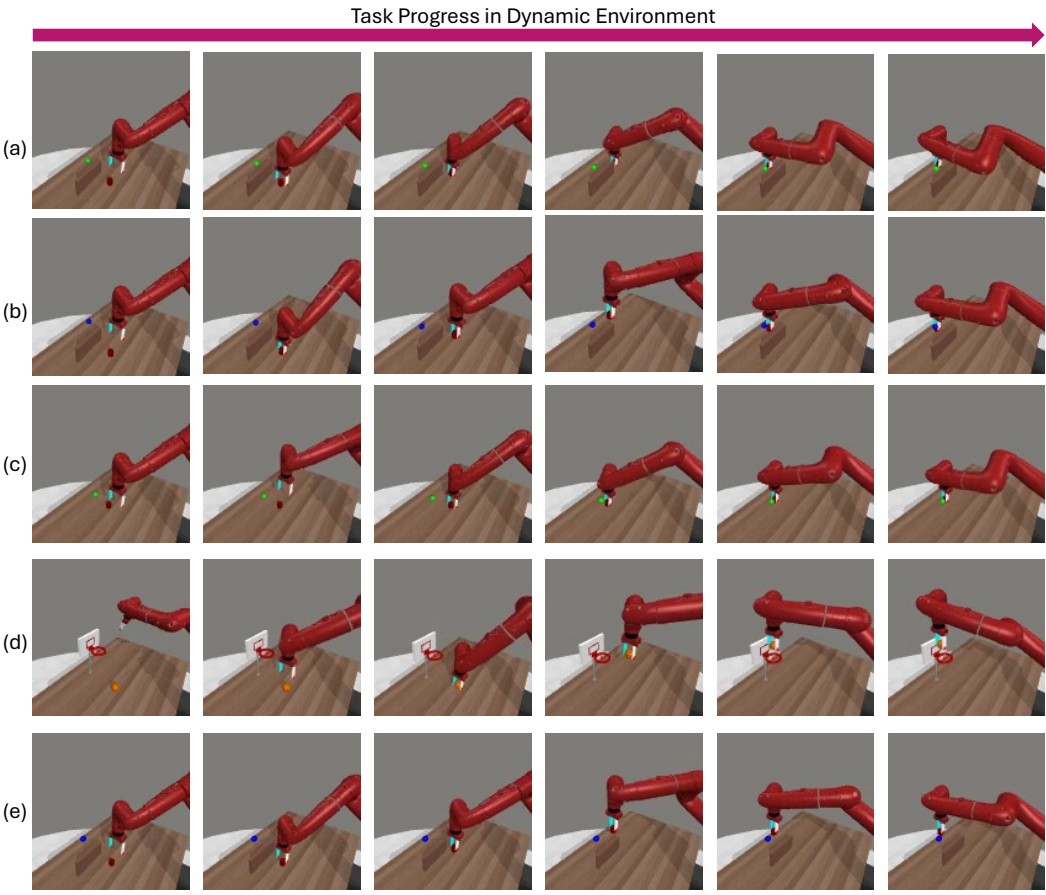

Figure 4: Qualitative results in Dynamic environments

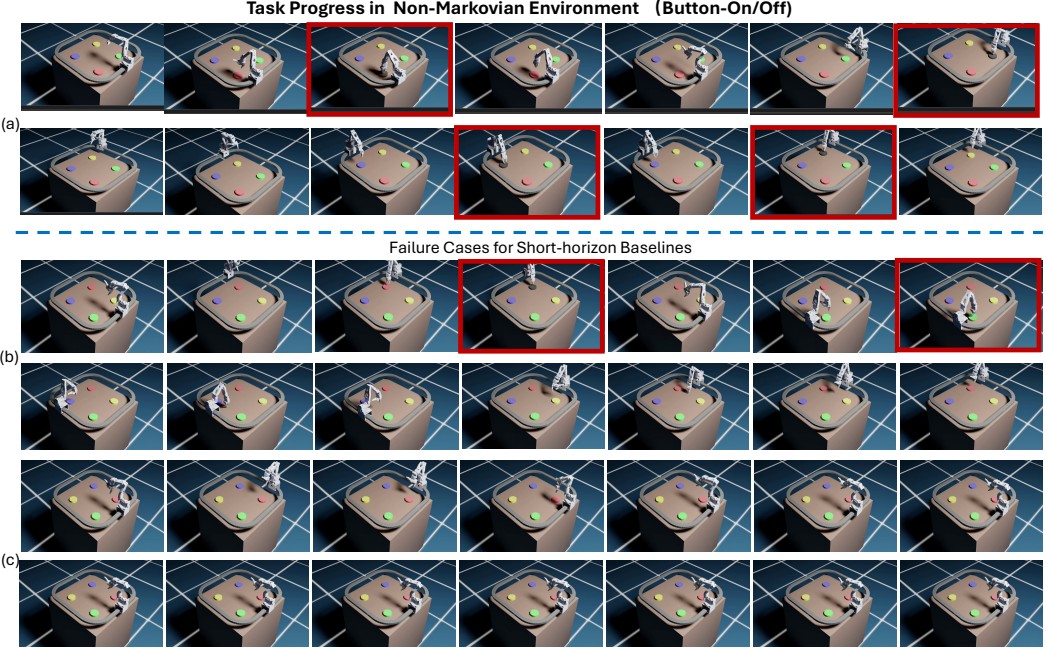

Figure 5: Qualitative results of Button Press On/Off Tasks. (a) is the result of MGP-Long; (b) and (c) are results of Short-Horizon baselines.

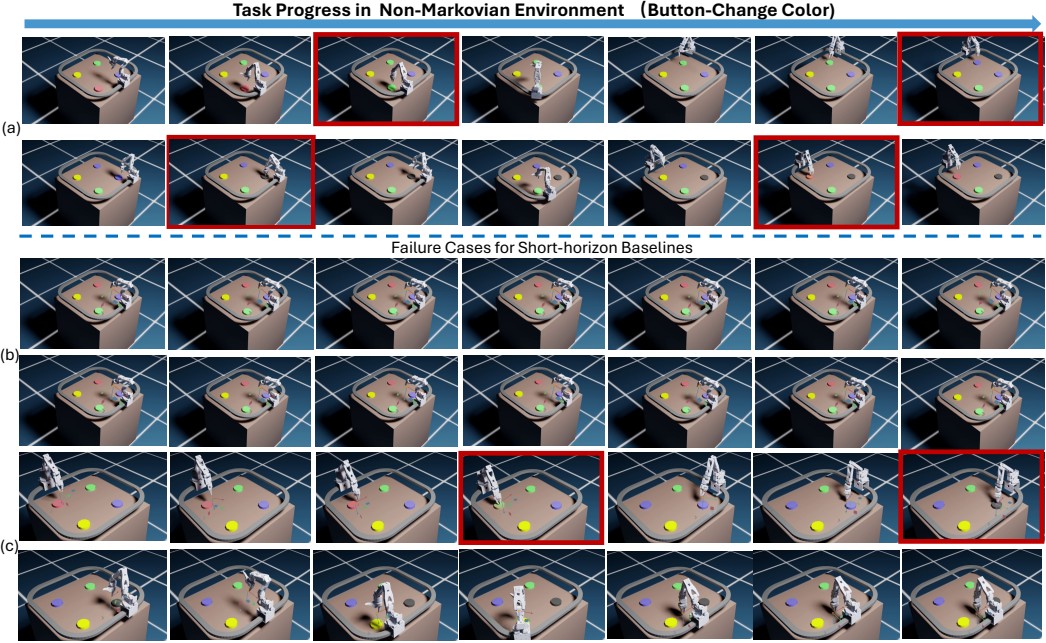

Figure 6: Qualitative results of Button Press Color Change Tasks. (a) is the result of MGP-Long; (b) and (c) are results of Short-Horizon baselines.

## C.5 CONFIDENCE SCORE ANALYSIS

To access how the confidence score behaves in practice, we plot token-level confidence over refinement passes as MGP-Long executes in Meta-World and in a dynamic environment.

Fig. 7 (a) shows a confidence heatmap for MetaWorld–Disassemble (quasi-static, 'Very Hard'). Fig. 7 (b) shows the corresponding mask–unmask map (black = masked at the first refinement, gray = masked at the second; the pink band marks the already-executed prefix). The horizontal axis corresponds to steps at which replanning occurs; after each step, 12 actions, corresponding to three tokens, are executed before replanning again. Two refinement passes occur at each replanning step. The vertical axis indexes temporal action tokens. Fig. 7(c) shows the rollout, with frequently edited action segments highlighted in red. The model is confident during approach motions but its confidence declines at grasp-and-lift, particularly when the first grasp fails and requires multiple corrections.

In order to investigate confidence under distribution shift, we also visualize how the confidence score behaves when the environment changes mid-episode. We consider a dynamic basketball task in which the scene is static at the start, and the hoop begins moving continuously halfway through the rollout. Fig. 8(a) shows the per-token confidence heatmap; Fig. 8(b) shows the corresponding mask–unmask map; Fig. 8(c) illustrates the rollout, with actions frequently edited highlighted in red; these coincide with the onset of hoop motion. Across these views, confidence drops and masking clusters around the transition to motion, which indicates that the policy focuses edits exactly where the dynamics shift.

We also measure the fraction of high-confidence vs low-confidence tokens that the model updates with a different value if prompted to resample them, as well as the impact of resampling high instead of low confidence tokens. On MetaWorld–Disassemble, at each replan we either (i) mask the bottom 70% (default) or (ii) mask the top 70% high-confidence tokens, and report task success and the flip rate (fraction of masked tokens that change after refinement). With low-confidence masking the policy reaches 0.86 success with a 60.6% flip rate—the model frequently edits tokens it was least confident about. With high-confidence masking, the flip rate falls to 15.1%, showing the model mostly keeps high-confidence tokens unchanged even when forced to revisit them. This indicates the confidence scores are probabilistically well calibrated.

## C.6 MODEL SIZE AND SPEED

Although training proceeds in two stages, the system is lightweight in practice: our policy has $\sim$7M parameters vs. $\sim$262M for DP3 (˜37× fewer), and training for 2000 epochs takes $\sim$55 minutes vs. $\sim$3.3 hours under the same setup. Inference is also lighter: diffusion policies require many iterative denoising steps, whereas our masked-generation planner predicts in parallel and performs only one or two refinement passes, yielding substantially lower deployment latency. Thus, the two-stage design does not increase runtime complexity and, in our experiments, is faster to train and deploy.

## C.7 ACCURACY OF TOKENIZED ACTIONS

To assess any inaccuracies due to mismatch between discrete tokens and continuous actions, we directly measured VQ-VAE reconstruction quality. On MetaWorld–Disassemble, the tokenizer's average per-step L2 reconstruction error is $\approx 1 \times 10^{-4}$; visual overlays of ground-truth and reconstructed actions are indistinguishable. Moreover, replacing ground-truth actions with their VQ-VAE reconstructions and replaying them yields 100% task success in MetaWorld. These results indicate that quantization error is negligible for control in our setting.

## C.8 MULTIMODALITY ANALYSIS

The Gumbel-Max trick ensures that token sampling still follows the logits' softmax probability distribution, preserving diversity in the sampled results. To verify this, we conducted a qualitative evaluation of our model on the Push-T task Florence et al. (2021). In the initial stage, the control point, T-block, and target are placed symmetrically in fixed locations. MGP-Short then autoregressively generates 16 subsequent actions (4 tokens) and executes the first 8 actions based on the

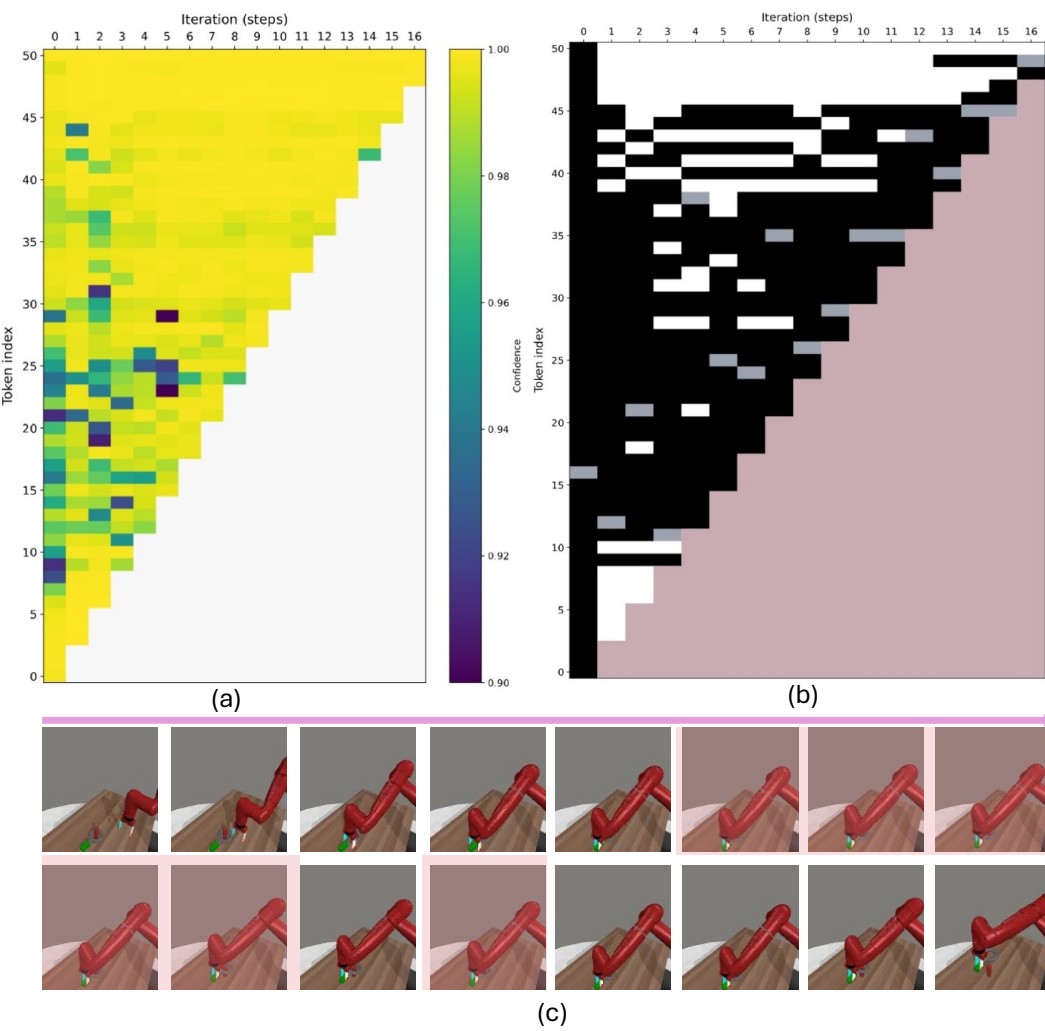

Figure 7: **Confidence score in MetaWorld 'Disassemble' environment.** (a) Per-token confidence heatmap across refinement iterations. (b) Mask–unmask map (black = masked at first refinement; gray = masked at second; pink = executed prefix). (c) Rollout snapshots with frequently edited segments highlighted. Confidence remains high during approach motions, then drops at fine, outcome-critical manipulations when the gripper must accurately grasp and lift the ring. These low-confidence tokens are repeatedly masked and corrected, showing that refinement concentrates edits where they matter.

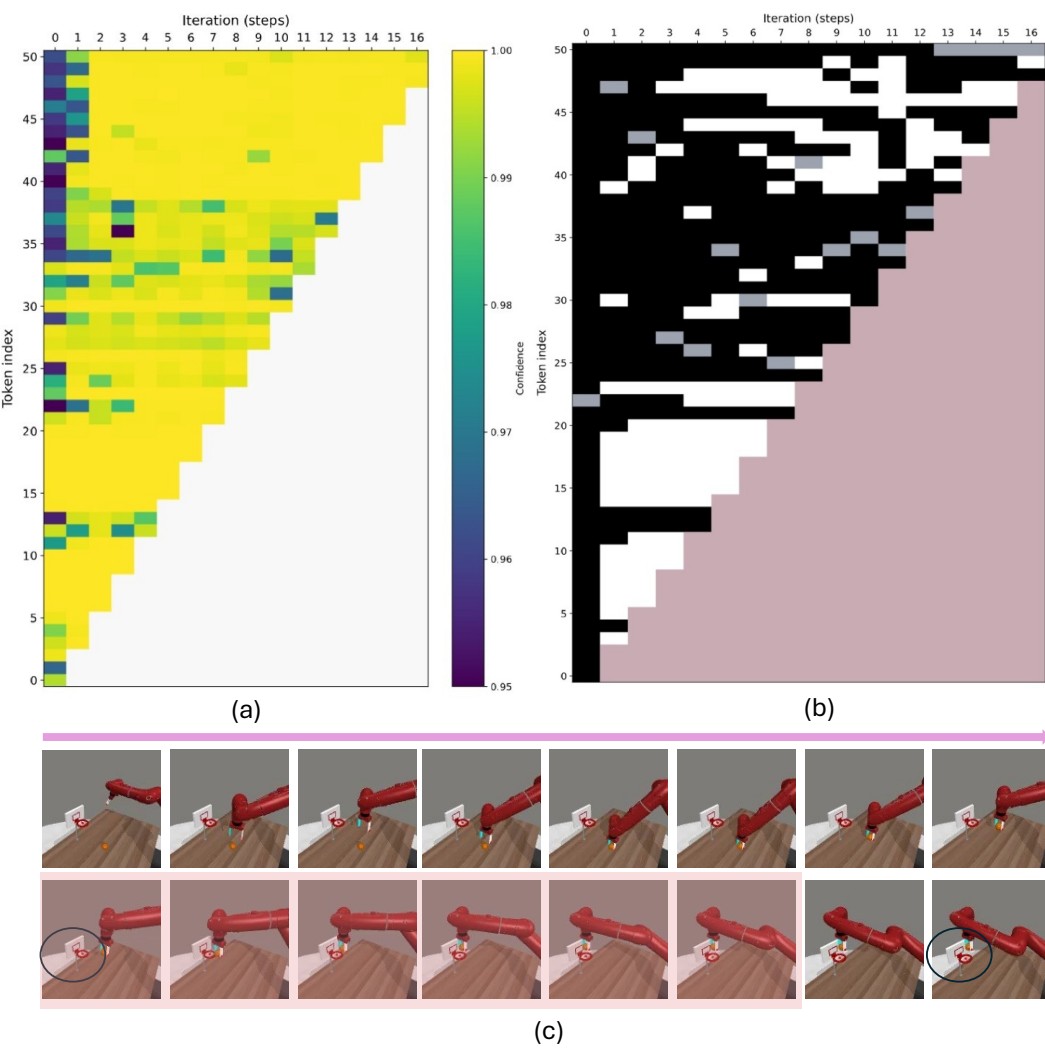

Figure 8: **Confidence under distribution shift in *Dynamic* 'Basketball' environment.** (a) Per-token confidence heatmap across refinement iterations. (b) Mask–unmask map (black = masked at first refinement; gray = masked at second; pink = executed prefix). (c) Rollout snapshots with frequently edited action segments highlighted. The hoop begins moving midway through the episode; confidence drops and masks cluster around the onset of motion, indicating that edits occur precisely when the dynamics change.

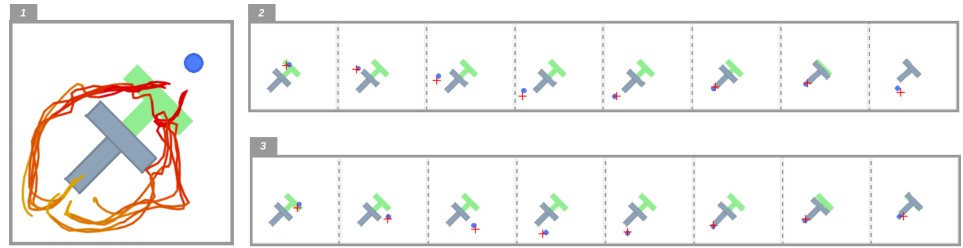

Figure 9: 1: The control point trajectories of the first 40 frames from 20 successful episodes. 2. An example of pushing the T-block from above. 3. An example of pushing the T-block from below.

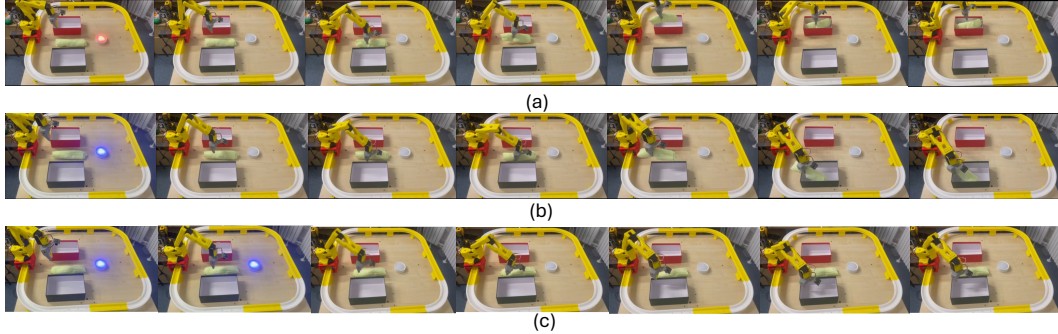

Figure 10: Qualitative results of MGP-Long on towel-sorting tasks. (a) is the results when the initial light is red; (b) is the results when the initial light is blue. (c) Failure case of DP3-Full Seq. on towel-sorting tasks.

previous frame's image and the control point position. The resulting trajectory plots (Fig. 9) show that MGP's sampling still preserves the diversity distribution of the original data.

## D  REAL-WORLD EXPERIMENTS

### D.1  IMPLEMENTATION DETAILS

We collect 60 human demonstrations. Each demonstration contains synchronized RGB images, depth maps, and colored point clouds, along with robot proprioception and end-effector (EE) signals. For each demonstration, we record approximately 100 timesteps. Point clouds are first cropped to the workspace and then downsampled to 1,024 points. Inputs are categorized into (1) observations and (2) robot state. We use downsampled point cloud as observations. Robot state, used as an additional conditioning input, includes joint states, EE position, EE orientation, and gripper state. The model outputs an action at each timestep consisting of the target EE position, EE orientation, and a gripper state.

### D.2  QUALITATIVE RESULTS

The qualitative results of MGP-Long on the towel-sorting task with different initial light colors are shown in Fig. 10 (a) and (b). We can clearly see that, even though the light switches off immediately after turning on, the robot still places the towel into the correct basket. In contrast, short-horizon baselines fail, as they cannot infer the correct basket from the available observations. Fig. 10 (c) also illustrates a typical failure case of DP3-FullSeq, where the robot fails to pick up the towel at the start.

We further investigate which inputs are most critical for performance. We find that the end-effector position is important, while the joint states have comparatively less influence. Visual observations (point clouds) play a major role: when we introduce a light color not seen in the training data (e.g., yellow), the robot still executes a plausible motion but fails to grasp the towel (it assumes the towel

is further to the left) and attempts to place it in the left basket. When we completely remove the point-cloud input, the policy becomes confused and simply hovers above the towel without taking action.

## E  USE OF LARGE LANGUAGE MODELS (LLMS)

Portions of the writing in this paper were polished using large language models.

