# OpenReview forum: "Masked Generative Policy for Robotic Control"
_ICLR.cc/2026/Conference — ICLR 2026 Poster_

### Official Review · Reviewer_wnrp · 2025-10-26

**Soundness:** 3
**Presentation:** 3
**Contribution:** 3
**Rating:** 8
**Confidence:** 3

**Summary:**

The paper introduces Masked Generative Policy, which is a new framework for visuomotor imitation learning that models robot actions as discrete tokens and leverages masked generative transformers to efficiently generate and refine action sequences. Unlike autoregressive or diffusion generative policies, MGP tries to generate globally coherent future plans and refine them online. It combines MaskGIT-style generation with robotic action modeling. The experimental results demonstrate state-of-the-art performance on Markovian and non-Markovian control.

**Strengths:**

- It reframes the policy generation problem as masked generative modeling is new and practical, especially given the latency and horizon challenges in robotics. The tokenization of actions is smart to allow transformer modeling of full sequences.
- The global coherence maintains long-horizon consistency through token memory. The parallel sampling and selective refinement drastically cut latency, leading to high inference efficiency.
- The experimental results are comprehensive and demonstrate the effectiveness of the proposed method across simulations and tasks. While diffusion models model smooth distributions and autoregressive models enforce causality, MGP smartly bridges them using mask-and-refine semantics, achieving both speed and robustness.

**Weaknesses:**

- The system design and two-stage training are complex. The VQ-VAE and MGT pipeline introduces extra overhead and possible distribution shift between discrete tokens and true continuous actions.
- When predicting all tokens at once, it loses the explicit notion of conditioning the next tokens on the current action. In dynamic control, this can lead to physically inconsistent predictions.
- The model must have enough context to predict consistent future tokens without sequential conditioning. It could work in structured simulation, but may fail with partial observability or noisy real-world sensors where causality exists.

**Questions:**

- Is the pipeline easy to smoothly transfer to real-world tasks? The robustness to sensor noise, delays, or physical contact uncertainty remains a question, especially when it requires a strong encoder and global context.
- There were few visual rollouts or per-task failure analyses. How token refinement behaves in specific dynamic scenes could be more illustrative.
- A discussion section on the potential domain mismatch and increased complexity of the proposed two-stage training is helpful.

---

> ### Author Response · Authors · 2025-11-21
> **Reply to reviewer wnrp (1/2)**
>
> Thanks for the positive reception of our work! We appreciate the detailed and helpful comments, and provide clarifications and responses to each point below.
>
> > Is the pipeline easy to smoothly transfer to real-world tasks? The robustness to sensor noise, delays, or physical contact uncertainty remains a question, especially when it requires a strong encoder and global context.
>
> We are currently conducting some non-Markovian real-world experiments with our method, and will update both this response and the paper with quantitative and qualitative results next week. Specifically, the setup consists of a light turned on in either red or yellow, after which it is switched off. A robot is then tasked with grasping a towel and placing it into one of two baskets based solely on the initial light color. Because the light is only visible at the very start of the episode, the task is non-Markovian, as later observations do not contain information about the required goal. Regarding partial observability, our simulation results in the missing-observation setting (Sec. 4.2 and Appendix Sec. C.2) already show that MGP-Long remains stable when sensor observations are intermittently unavailable, even if the probability of observation missing reaches 70%.  We will also include scenarios involving missing observations in real-world experiments.
>
> > There were few visual rollouts or per-task failure analyses. How token refinement behaves in specific dynamic scenes could be more illustrative.
>
> In the supplementary material, we already include around five successful visual rollouts per dynamic environment and three successful and unsuccessful visual rollouts per non-Markovian environment for MGP-Long (our methods) and short-horizon baselines. More visual rollouts have been updated in the supplementary materials. Some reasons for failures of non-Markovian environments are discussed in Appendix C.4: short-horizon baselines often stall or press the wrong buttons because they lack global trajectories modeling. More detailed failure analyses are given in Appendix C.3 and C.4.
>
> For token refinement in dynamic environments, we now visualise how token confidence scores change across refinement iterations as actions are executed (Fig. 8). For MGP-Long in a dynamic basketball environment, confidence is high while the hoop is stationary and drops once the hoop begins moving. The visualizations show that drops in confidence align with environmental changes. Additional figures and discussion are provided in Sec. 4.6 and Appendix C.5 of the updated paper.
>
> > A discussion section on the potential domain mismatch and increased complexity of the proposed two-stage training is helpful. The system design and two-stage training are complex. The VQ-VAE and MGT pipeline introduces extra overhead and possible distribution shift between discrete tokens and true continuous actions.
>
> Thanks for raising this! We agree it merits explicit discussion, and we have updated the discussion about the complexity of two-stage training and performance of VQ-VAE (See Appendix C.6 and C.7). As for the potential domain mismatch between discrete tokens and true continuous actions, our VQ reconstruction error is extremely low: such as In MetaWorld Disassemble task, the VQ tokenizer’s average per-step L2 reconstruction error is ≈ 1e-4; visual overlays of ground-truth vs. reconstructed actions are indistinguishable. Moreover, replacing ground-truth actions with their VQ reconstructions and replaying them yields 100% task success in MetaWorld. Together, these results indicate that quantization error is negligible for control in our setting.
>
> As for introducing extra overhead, we compare our model complexity, inference process, model runtime (and convergence time) with DP3. We found that although training proceeds in two stages, the system is very lightweight in practice: our policy has ~7M parameters vs. ~262M for DP3, about 37× fewer parameters, and training for 2000 epochs takes ~55 min vs. ~3h under the same setting (See Section 4.1 and Appendix C.6). Inference is also very light: while diffusion policies incur iterative sampling, our masked-generation planner performs parallel prediction followed by only one or two refinement steps, giving significantly lower deployment latency. Thus, the two-stage design does not increase runtime complexity and is, in our experiments, faster to train and to deploy.

---

> ### Author Response · Authors · 2025-11-21
> **Reply to reviewer wnrp (2/2)**
>
> > When predicting all tokens at once, it loses the explicit notion of conditioning the next tokens on the current action. In dynamic control, this can lead to physically inconsistent predictions.
>
> Thanks for raising this concern. In practice, MGP-Long minimizes open-loop exposure: Although it predicts a global trajectory once, it only executes a short chunk before replanning from the latest observation using posterior-confidence editing. In other words, the model conditions on the actually executed actions at every step, and the effective open-loop horizon is tightly bound by the executed action window. This design preserves physical consistency for three reasons: (1) All future predictions are conditioned on the executed actions and the current observation. (2) Because we execute at most 2 tokens (for MetaWorld) before re-evaluation, any mismatch can only propagate within a small window. (3) Confidence-guided masking edits only low-confidence regions of the action tokens, while high-confidence (physically consistent) tokens remain frozen while maintaining stability.
>
> We believe we’ve covered all points raised; please let us know if anything else would be helpful.

---

> > ### Comment · Reviewer_wnrp · 2025-11-27
> > **Reply to Submission6226 Authors**
> >
> > Thank the authors for the detailed response. I have no major concerns and will keep my positive recommendation. I would love to discuss with the other reviewers.

---

> ### Author Response · Authors · 2025-11-27
> **Additional Reply to reviewer wnrp**
>
> > Is the pipeline easy to smoothly transfer to real-world tasks? The robustness to sensor noise, delays, or physical contact uncertainty remains a question, especially when it requires a strong encoder and global context.
>
> We have now deployed our model operating a LeRobot arm on a real world non-Markovian towel-sorting task. A towel and two baskets are placed in the robot's working area with a random lateral offset. At the beginning of each episode, a light briefly turns on in either red or blue and is then switched off; the color indicates which basket the towel should be placed in. The robot must grasp a towel and place it into the correct basket based solely on the initial light color. As the light is only visible at the very start of the episode, later observations do not contain any information about the desired goal, making the task non-Markovian. We collect 60 expert demonstrations by teleoperation of the LeRobot arm. For each trial, we record robot joint positions, end-effector position and orientation, gripper state, and synchronized RGB, depth, and point-cloud data. We evaluate our model in 25 real-world trials. MGP-Long achieves a success rate of 96%, outperforming DP3-Full Seq (which achieves 84%). Furthermore, MGP-Long maintains the same success rate even when observations are intentionally dropped during action execution. These results demonstrate that our pipeline transfers smoothly to real-world settings and that MGP-Long’s global reasoning allows it to handle noisy, partial, and missing observations without degrading performance. Please see Sec. 5 and Appendix Sec. D in the updated paper for details.

---

### Official Review · Reviewer_4ZDe · 2025-10-28

**Soundness:** 3
**Presentation:** 3
**Contribution:** 2
**Rating:** 6
**Confidence:** 3

**Summary:**

This paper introduces Masked Generative Policy (MGP), a new visuomotor imitation learning framework that models robot control as a masked token-generation problem.
MGP first discretizes continuous actions with a VQ-VAE tokenizer, then trains a masked generative transformer (MGT) to reconstruct full action sequences from partially masked tokens conditioned on current observations.

Two inference paradigms are proposed:

- MGP-Short for Markovian, short-horizon tasks: parallel token generation with one or two score-based refinement steps.

- MGP-Long for non-Markovian, long-horizon tasks: predicts the entire trajectory in one pass and adaptively refines uncertain future tokens through posterior-confidence estimation (PCE) as new observations arrive.

Extensive experiments on Meta-World and LIBERO benchmarks show strong gains—up to 35× faster inference and higher success rates (+9% overall, +60% in dynamic or missing-observation settings).
Ablations (MGP-FullSeq, MGP-w/o-SM) validate that PCE-based selective refinement is critical for efficiency and global coherence.

**Strengths:**

Original idea: creatively transfers masked-generation paradigms (MaskGIT/MUSE) to robotic action synthesis.

Technical soundness: clearly defined VQ-VAE tokenizer, transformer conditioning, and confidence-guided refinement loop.

Empirical rigor: evaluated on 150+ tasks across difficulty levels; includes robustness tests (dynamic, missing-observation, non-Markovian).

Fair comparison: benchmarks against continuous-action (diffusion/flow) and discrete-token baselines under identical encoders and demos.

Ablation insight: MGP-w/o-SM (without score-based masking) confirms that selective refinement improves both efficiency and success rate.

Relevance: unifies the advantages of diffusion (sample quality) and autoregressive (temporal coherence) methods in a parallelizable design.

**Weaknesses:**

Limited analysis of tokenizer sensitivity: performance may depend on the VQ-VAE codebook design, but this is not explored.

Hyperparameter transparency: the exact confidence-masking threshold and its effect on refinement stability are not analyzed.

Potential complexity: the two-stage training (tokenizer + policy) increases implementation effort; joint end-to-end training would strengthen the approach.

**Questions:**

How is the confidence-based masking threshold determined? Fixed ratio or adaptive per step?

Does the posterior-confidence estimation ever over-mask or destabilize refinement when confidence calibration drifts?

How sensitive is performance to the tokenizer’s codebook size and discretization granularity?

Would an end-to-end jointly trained transformer + VQ-VAE outperform the current two-stage pipeline?

Discrete tokens normally introduce information loss—what do the authors believe enables MGP’s discrete representation to outperform continuous-action models like Diffusion Policy? Is it the global trajectory modeling, masked refinement dynamics, or some property of the VQ-VAE discretization?

---

> ### Author Response · Authors · 2025-11-21
> **Reply to reviewer 4ZDe (1/3)**
>
> We are grateful to the reviewer 4ZDe for the thorough and insightful comments. These suggestions have helped us improve the clarity of the work, and we address each point in detail below.
>
> > How is the confidence-based masking threshold determined? Fixed ratio or adaptive per step? Hyperparameter transparency: the exact confidence-masking threshold and its effect on refinement stability are not analyzed.
>
> Thank you for the suggestion. We currently use a fixed-ratio masking rule: at each replan, we recompute posteriors on the unexecuted suffix, rank tokens by confidence, and mask the bottom 70% for one refinement pass. We ablated the masking ratio (50%, 70%, 85%) for MGP-Long on five Meta-World Very Hard tasks. 70% achieves the best average success rate on both settings. A 50% ratio underperforms because leaving too many low-confidence tokens unedited allows weak tokens to persist and can also interfere with accurately regenerating the masked ones. The result for 85% is comparable to 70% on average. We have added the results and discussion to the updated paper (Sec. 4.5 and Appx. B.5; Table 13). We agree that exploring an adaptive mask ratio would be an interesting direction for future work.
>
> | Mask Ratio  | Shelf place | Disassemble| Stick pull | Stick push| Pick place wall| Average |
> |------------------      |-------------|-------------|------------|------------|------------------|---------|
> | 50%  | 0.25        | 0.83        | 0.41       | **1.00**       | 0.31             | 0.560   |
> |70%  |**0.29**        | **0.86**    | **0.45**       | **1.00**       | **0.33**            | **0.586**   |
> | 85% | 0.27        | 0.85        | 0.45       | **1.00**       | 0.31             | 0.576   |
>
> > Does the posterior-confidence estimation ever over-mask or destabilize refinement when confidence calibration drifts?
>
> In terms of the overmasking problem, in our mask-ratio ablation (50%, 70%, 85%) for MGP-Long, task success remains stable even at 85%, with 70% giving the best average. If over-masking were occurring, we would expect a clear drop at high ratios; we do not observe that (Sec. 4.5 and Appx. B.5; Table 13).
>
> In terms of the calibration drift and refinement stability, posterior confidence estimation in Adaptive Token Refinement (ATR) mechanism, keep refinement well-behaved. First, at every replan we recompute posteriors from the latest observation and the already-executed actions, rather than reusing stale scores. In ablations, using fresh posteriors (ATR) outperforms Score-Reuse by +5.53% and Random by +10.68% on Meta-World Hard/Very-Hard, indicating that fresh posteriors are crucial for stability (see paper, Sec. 4.5 and Appendix B.6). Second, ATR executes only a short token chunk, anchors it, then re-evaluates the remaining tokens with the new observation, editing low-confidence positions. This bounds error propagation and provides frequent opportunities to correct drift. Concretely, on Meta-World Disassemble we deliberately perturbed the first four high-confidence tokens after the initial global prediction and then ran ATR; success dropped modestly from 0.86 → 0.80 (Δ −0.06), indicating that refinement remains stable and recovers much of the performance even with injected early errors.
>
> > Would an end-to-end jointly trained transformer + VQ-VAE outperform the current two-stage pipeline? Potential complexity: the two-stage training (tokenizer + policy) increases implementation effort; joint end-to-end training would strengthen the approach.
>
> Thank you for the interesting idea. Currently, we follow the standard two-stage training paradigm, which has been widely shown to be effective and robust across different domains:
> MaskGIT (Huiwen Chang,CVPR’22), MAGVIT (Lijun Yu,CVPR’23)/MAGVIT-v2 for video, and recent discrete-token policy models such as VQ-BeT (Muhammad Shafiullah,ICML’24), VQ-VLA (Yating Wang,ICCV’25), MMM (Ekkasit Pinyoanuntapong,CVPR’25), and Quest (Mete,NeurIPS’24). Joint end-to-end training is an interesting direction that could simplify the training pipeline, and we will explore its feasibility in future work.

---

> ### Author Response · Authors · 2025-11-21
> **Reply to reviewer 4ZDe (2/3)**
>
> > How sensitive is performance to the tokenizer’s codebook size and discretization granularity? Limited analysis of tokenizer sensitivity: performance may depend on the VQ-VAE codebook design, but this is not explored.
>
> We agree it is important to analyze sensitivity to tokenizer hyperparameters, and have now run two additional sets of experiments.
> Codebook size. We adjust the VQ-VAE codebook size for MGP-Short on the five Meta-World Very Hard tasks, comparing 512, 1024, and 2048 entries. The average success rates are 0.534, 0.538, and 0.522, respectively. The differences are small (≤ 1.6 pp) and show no consistent trend, indicating that performance in this setting is not particularly sensitive to codebook size. We therefore keep 1024 as the default, as it is slightly better while offering a good capacity/efficiency trade-off. We have incorporated these results and discussion into the updated paper (Sec. 4.5 and Appx. B.2; Table 10).
>
> | Codebook Size  | Shelf place | Disassemble| Stick pull | Stick push| Pick place wall| Average |
> |------------------     |-------------|-------------|------------|------------|------------------|---------|
> |512    | **0.23**        | **0.81**       | 0.41       | 0.87       | 0.35             | 0.534   |
> |1024  | 0.20        | 0.74       | **0.50**       | **0.90**       | 0.35             | **0.538**|
> | 2048 | 0.21        | 0.80       | 0.39       | 0.85       | **0.36**             | 0.522   |
>
> Discretization granularity:  Our VQ-VAE tokenizer converts every 4 consecutive primitive actions into one discrete token (for both MGP-Short and MGP-Long). We have added an experiment varying discretization granularity by testing 2 actions/token and 8 actions/token on the five Meta-World Very Hard tasks. Averaged over tasks, 4 actions/token achieves the highest success rate (0.538), outperforming 2 actions/token (0.526, +1.2 pp) and 8 actions/token (0.514, +2.4 pp). Thus, changes in token granularity have limited effect on final task performance. We have incorporated these results and discussion into the updated paper (Sec. 4.5 and Appx. B.3; Table 11).
>
> | Discretization Granularity   | Shelf place | Disassemble | Stick pull | Stick push | Pick place wall | Average |
> |------------------      |-------------|-------------|------------|------------|------------------|---------|
> | 2 actions/token  | 0.21        | **0.90**        | 0.33       | 0.88       | 0.31             | 0.526   |
> | 4 actions/token  | 0.20        | 0.74        | **0.50**       | **0.90**       | **0.35**             | **0.538**   |
> | 8 actions/token  | **0.25**        | 0.78        | 0.35       | 0.86       | 0.33             | 0.514   |

---

> ### Author Response · Authors · 2025-11-21
> **Reply to reviewer 4ZDe (3/3)**
>
> > Discrete tokens normally introduce information loss—what do the authors believe enables MGP’s discrete representation to outperform continuous-action models like Diffusion Policy? Is it the global trajectory modeling, masked refinement dynamics, or some property of the VQ-VAE discretization?
>
> Our advantage comes from a combination of (A) high-fidelity discretization (so tokenization itself does not significantly degrade actions), plus (B) global trajectory modeling with confidence-guided masked refinement that continuous step-by-step generators typically lack.
>
> Masked generative transformers (MGT) are designed to operate on discrete tokens: they predict all tokens in parallel and then refine masked action tokens. Prior work shows that this yields high fidelity with far lower decoding cost than sequential autoregression or iterative diffusion (e.g., MaskGIT (CVPR’22)). Recent visual tokenization results (e.g., MAGVIT-v2’s “Tokenizer is key” finding) further indicate that with a strong tokenizer, next-token language models can match or even surpass diffusion on standard image/video benchmarks, supporting the premise that high-quality discrete codes do not force a fidelity trade-off. Recent VQ-based robot policies show that mapping continuous actions into a compact discrete code space improves multimodality, planning efficiency, and yields reusable/sharable representations (e.g., VQ-BeT (ICML’24); Quest (NeurIPS’24)). Scaling vector-quantized action tokenizers further improves long-horizon performance, suggesting that stronger codebooks better capture the structure of feasible actions (VQ-VLA (ICCV’25)). In human-motion generation, works such as MMM (CVPR’25) demonstrate that masked generation over discrete codes achieves high-fidelity sequences with efficient refinement and reduced jitter.
>
> In our experiments, the VQ tokenizer’s reconstruction error is extremely low (e.g., on the Disassemble task in the MetaWorld, the average per-step L2 error is ≈1e-4). Visual overlays of ground-truth vs. reconstructed actions are indistinguishable. We also ran a “VQ reconstruction” check: replacing ground-truth actions with their VQ-reconstructed counterparts and replaying them in the environment yields 100% task success in MetaWorld. This indicates that quantization error is negligible for control in our setting, i.e., the discrete representation retains the fidelity needed for precise manipulation.
>
> Besides high-fidelity discretization, MGP benefits from global planning with dynamic refinement, and parallel generation (thus low latency). MGP-Long proposes an entire trajectory in one shot, then refines only low-confidence unexecuted tokens using the latest observation (adaptive token refinement, ATR) (See Sec.3.4). This preserves global coherence while adapting locally when the world changes mid-episode. Diffusion policies typically generate short windows with receding horizons, which can drift or lose long-range structure under non-stationarity. Masked token generation is parallel and needs only one or two refinement passes, avoiding the many denoising steps of diffusion at test time which matters for real-time control loops.
>
> We believe we have addressed all your comments; please let us know if anything else would be useful.

---

### Official Review · Reviewer_UA2E · 2025-10-29

**Soundness:** 3
**Presentation:** 2
**Contribution:** 3
**Rating:** 6
**Confidence:** 3

**Summary:**

This manuscript proposes a novel imitation learning framework for learning visuomotor policy parameterized by masked generative transformer (MGT), which enables high inference efficiency for closed-loop control while maintaining robustness in long-horizon and non-Markovian tasks. Specifically, two sampling strategies are designed: (1) MGP-Short performs short-horizon sampling and refines action tokens with few iterations for the best performance-efficiency trade-off in Markovian tasks; and (2) MGP-Long samples the full trajectory and adaptively refines tokens with updated observations from the environment to retain global coherence. Experiments demonstrate the strong performance of the proposed methods in Markovian and more challenging tasks.

**Strengths:**

- Unlike diffusion-based policy, which might require external distillation for fast inference speed, MGP puts less stress on iterative sampling for obtaining clean actions, and has high flexibility of test-time adjustment with proposed sampling strategies.
- MGP-Long iteratively refines the action tokens using the executed actions along with the updated observation to improve trajectory-level coherence, which achieves strong performance in Non-Markovian and dynamic environments, and remains robust to missing observations

**Weaknesses:**

- Baselines such as diffusion-based policies (e.g. ) as well as VQ-BeT stand out when learning multimodal action distributions, while MGP is also built on top of vector quantization, it is not yet clear how the proposed sampling methods work on tasks with explicit multimodality
- As all tokens are predicted in parallel, the refinement process can be affected if there are low-quality actions predicted initially with high confidence, causing error accumulation throughout the following iterations. Furthermore, it would be helpful to extend the first ablation studies to investigate how many performance gains can be obtained from more refinement steps, especially in more challenging environments.
- Please include standard deviations in the table for thoroughness if multiple seeds are used to aggregate the result.

**Questions:**

- Typo: “blcoks” -> “blocks” in line 191
- In Figure 3, should the unexecuted token “52” at the bottom left be “53” before Posterior-Confidence Estimation
- In line 269, the authors mentioned four ablation studies were conducted, but in section 4.5, only three of them are elaborated.
- How many actions are encoded into one discrete token? And would that hyperparameter affect performance on different tasks?

---

> ### Author Response · Authors · 2025-11-21
> **Reply to reviewer UA2E (1/2)**
>
> Thanks for the positive reception of the paper, and the detailed, constructive feedback. We have carefully considered the comments and addressed the raised points below.
> > Baselines such as diffusion-based policies (e.g. ) as well as VQ-BeT stand out when learning multimodal action distributions, while MGP is also built on top of vector quantization, it is not yet clear how the proposed sampling methods work on tasks with explicit multimodality.
>
> We agree that explicitly characterizing multimodality is important. Below we clarify how MGP handles multimodal action distributions in practice and provide an additional experiment that directly probes this property:
>
> 1. Due to the discretization, action token generation becomes a process of sampling from a categorical distribution in general. For MGP, we adopt the Gumbel-Max trick for sampling, where we first generate Gumbel noise to be added to the logits. Gumbel noise is mostly distributed around smaller values, but it also has a certain probability of producing very large values. After adding Gumbel noise, the occasional extreme values from the random Gumbel noise allow otherwise low-probability logits to 'win' the argmax, producing diversity in the results. (See Sec. 3.3)
>
> 2. We conducted an experiment on the Push-T environment (Pete Florence, CoRL’21) and showed the diversity samples from our model given a fixed symmetric initialization. Our model will predict the following 16 actions and execute the first 8 actions. The 20 trajectory samples of the successful episode confirmed the sampling diversity of our model. The details of the Push-T experiment have been added to Appendix Sec C.8. of the updated paper, and visualizations of the multimodal action distributions are provided in Figure 9.
>
> > As all tokens are predicted in parallel, the refinement process can be affected if there are low-quality actions predicted initially with high confidence, causing error accumulation throughout the following iterations.
>
> Thank you for the thoughtful comment. In MGP-Long, initial predictions do not accumulate errors because they are repeatedly re-evaluated in a closed loop. First, tokens are generated in parallel under the same context (Sec. 3.3), so later tokens are not conditioned on potentially erroneous early samples. Second, only a short chunk of action executed, then obtain a fresh observation and let ATR recompute posteriors for the unexecuted suffix and edit only low-confidence positions (Sec. 3.4). If an early token was confident yet suboptimal, the new observation reduces confidence on the affected future tokens; those tokens are selected for editing and the plan self-corrects before further execution. This closed-loop editability is not available in standard autoregressive rollouts.
>
> To make this concrete, we conducted an experiment on Meta-World Disassemble (“Very Hard”). After the initial global prediction, we deliberately corrupted the plan by overwriting the first four high-confidence tokens with random indices, then ran ATR normally.  The success rate with corruption was 0.80, versus 0.86 without (Δ −0.06). Despite injecting early errors, performance degrades only modestly. This supports our claim that MGP-Long’s closed-loop refinement has the potential to decrease error accumulation.
> We agree there can be extreme, adversarial cases where a single executed chunk is suboptimal; handling such cases better is an interesting direction for future work.
>
> > How many actions are encoded into one discrete token? And would that hyperparameter affect performance on different tasks?
>
> Our VQ-VAE tokenizer converts every 4 consecutive primitive actions into one discrete token (for both MGP-Short and MGP-Long). We have added an experiment varying discretization granularity by testing 2 actions/token and 8 actions/token on the five Meta-World Very Hard tasks. Averaged over tasks, 4 actions/token achieves the highest success rate (0.538), outperforming 2 actions/token (0.526, +1.2 pp) and 8 actions/token (0.514, +2.4 pp). Thus, changes in token granularity have a limited effect on final task performance. We have incorporated these results and discussion into the updated paper (Sec. 4.5 and Appx. B.3; Table 11).
>
> | Discretization Granularity   | Shelf place | Disassemble | Stick pull | Stick push | Pick place wall | Average |
> |------------------      |-------------|-------------|------------|------------|------------------|---------|
> | 2 actions/token  | 0.21        | **0.90**        | 0.33       | 0.88       | 0.31             | 0.526   |
> | 4 actions/token  | 0.20        | 0.74        | **0.50**       | **0.90**       | **0.35**             | **0.538**   |
> | 8 actions/token  | **0.25**        | 0.78        | 0.35       | 0.86       | 0.33             | 0.514   |

---

> ### Author Response · Authors · 2025-11-21
> **Reply to reviewer UA2E (2/2)**
>
> > Furthermore, it would be helpful to extend the first ablation studies to investigate how many performance gains can be obtained from more refinement steps, especially in more challenging environments.
>
> Thanks for the suggestion; we have now extended the first ablation study from MGP-Short to MGP-Long in dynamic environments. We evaluated r ∈ {1,2,3} on five dynamic tasks and reported the average success rate (SR). Results are given below (and also Table 12 in the revised paper). From the table we can see, increasing the refinement step from r=1 to r=2 improves SR by +5.2%, indicating that the score-based masking scheme is helpful in more challenging settings. Increasing further from r=2 to r=3 yields <1% additional SR on average, while incurring higher inference cost (extra forward passes). Given this accuracy–latency trade-off, we adopt r=2 for mgp-long in all subsequent experiments. We have incorporated these results and discussion into the updated paper (Sec. 4.5 and Appx. B.4; Table 12).
>
> | Refinement Steps   | Basketball | Pick place wall(W) | Pick place wall(T) | Push| Push wall(T)| Average |
> |------------------      |-------------|-------------|------------|------------|------------------|---------|
> | 1  | **1.00**        | **0.31**        | 0.08       | 0.15       | 0.38             | 0.384   |
> |2  |**1.00**        | 0.30        | **0.13**       | **0.25**    | **0.50**        | **0.436**   |
> | 3 | **1.00**        | **0.31**        | 0.10       | 0.20       | 0.48             | 0.418   |
>
> > Typo issues
>
> Thanks for pointing out these minor issues; we have fixed them in the revised paper.
>
> We hope this fully addresses your comments—happy to clarify anything further.

---

### Official Review · Reviewer_x3HB · 2025-11-02

**Soundness:** 3
**Presentation:** 3
**Contribution:** 3
**Rating:** 6
**Confidence:** 4

**Summary:**

This paper introduces the Masked Generative Policy (MGP), a novel framework for robot imitation learning that eliminates the inference bottlenecks of diffusion models and the sequential constraints of autoregressive models.
MGP-Short is specifically designed for Markovian tasks, adapting the masked generative transformer for short-horizon sampling. It demonstrates improved success rates on standard benchmarks while significantly reducing inference time.
MGP-Long allows for globally coherent predictions over long horizons, enabling dynamic adaptation, robust execution under partial observability, and efficient, flexible execution. It achieves state-of-the-art results in dynamic, observation-missing, and non-Markovian long-duration environments.
The authors validated the effectiveness of MGP in multiple simulated environments.

**Strengths:**

The authors conducted a thorough analysis of current action generation methods and proposed MGP to address the latency issues inherent in diffusion-style or autoregressive-style action generation. The paper is clearly articulated and easy to follow. The concept of using MGP to re-predict tokens with low confidence while maintaining those with high confidence is intriguing. Theoretically, this approach could indeed reduce the time consumed in predicting actions.

**Weaknesses:**

1. I acknowledge that the results in the simulated environment are impressive. However, due to the sim-to-real gap, it is often necessary to demonstrate effectiveness in real-world settings within this field.

2. Regarding the confidence score. Could you analyze the situations that might lead to a lower confidence score? Additionally, how can we ensure the accuracy of the confidence score itself?

3. About the MGP-Long settings. In long sequences, certain objects may cause environmental changes due to previous actions. At this point, the predictions may no longer remain globally coherent, and we would need to generate a new action sequence based on the changed objects.

**Questions:**

1. The results in the simulated environment are impressive. However, due to the sim-to-real gap, it is often necessary to demonstrate effectiveness in real-world settings within this field.
2. How can we ensure the accuracy of the confidence score itself?
3. Regarding the MGP-Long settings: In lengthy sequences, some objects may lead to environmental changes as a result of prior actions. When this occurs, the predictions may lose their overall coherence, necessitating the generation of a new action sequence that takes into account the modified objects.

---

> ### Author Response · Authors · 2025-11-21
> **Reply to reviewer x3HB (1/2)**
>
> Thanks for the positive reception of our work, and the thoughtful and detailed comments. We have addressed the raised points below.
> > The results in the simulated environment are impressive. However, due to the sim-to-real gap, it is often necessary to demonstrate effectiveness in real-world settings.
>
> We agree that evaluating real-world performance is important. We are currently conducting some non-Markovian real-world experiments with our method, and will update both this response and the paper with quantitative and qualitative results next week. Specifically, the setup consists of a light turned on in either red or yellow, after which it is switched off. A robot is then tasked with grasping a towel and placing it into one of two baskets based solely on the initial light color. Because the light is only visible at the very start of the episode, the task is non-Markovian, as later observations do not contain information about the required goal. We will also include scenarios involving missing observations.
> > Could you analyze the situations that might lead to a lower confidence score? How can we ensure the accuracy of the confidence score itself?
>
> Thanks for the suggestions. We have added visualizations of per-token confidence across refinement iterations as actions are executed by MGP-Long, together with mask–unmask maps. Please see Fig. 7 and Fig. 8 in the revised paper. On MetaWorld Disassemble (quasi-static, “Very Hard”), confidence remains high during approach phases but becomes low during precise, outcome-critical manipulations and at actions requiring repeated attempts—for example, it stays high while the gripper approaches the ring, then falls when the gripper must position itself accurately to grasp and lift, especially when the first grasp fails and the policy makes repeated adjustments; these low-score tokens are repeatedly masked and corrected. On dynamic Basketball with a moving hoop, confidence is high while the hoop is stationary and drops once the hoop begins moving. Thus, drops in confidence typically align with environment changes, and that refinement focuses on edits exactly where they matter. Additional figures and discussion are provided in Sec. 4.6 and Appendix C.5 of the updated paper.
>
> Quantitatively, we already included an experiment changing how scores are computed in MGP-Long: transformer posteriors recomputed after each short execution from the latest observation and executed prefix versus (a) random scores and (b) reusing stale scores. Posterior-based scores consistently outperform both, indicating that recomputed posteriors provide the accurate, up-to-date uncertainty needed to select tokens for editing (see paper, Sec. 4.5 / Appx. B.6).
> We have added two further quantitative experiments to show the confidence scores give meaningful information.
>
> 1. We measure the effect of varying the mask ratio, i.e. what proportion of (low confidence) tokens are resampled. At each replan we rank the unexecuted tokens by posterior confidence and in our standard setting mask the bottom 70% for one refinement pass. Changing between 50%, 70%, and 85% shows that 70% yields the best average success  across five Meta-World Very Hard tasks. A 50% ratio underperforms because many low-confidence tokens remain unedited and can also interfere with accurate regeneration of the masked ones; 85% is comparable to 70% on average.(Sec. 4.5; Appx. B.5, Table 13).
> 2. We measure the fraction of high-confidence vs low-confidence tokens that the model updates with a different value if prompted to resample them, as well as the impact of resampling high instead of low confidence tokens. On MetaWorld–Disassemble, at each replan we either (i) mask the bottom 70% (default) or (ii) mask the top 70% high-confidence tokens, and report task success and the flip rate (fraction of masked tokens that change after refinement). With low-confidence masking the policy reaches 0.86 success with a 60.6% flip rate—the model frequently edits tokens it was least confident about. With high-confidence masking, flip rate falls to 15.1%, showing the model mostly keeps high-confidence tokens unchanged even when forced to revisit them. This indicates the confidence scores are probabilistically well calibrated. We have added this experiment and analysis to the updated paper (see Sec. 4.6 and Appendix C.5).

---

> ### Author Response · Authors · 2025-11-21
> **Reply to reviewer x3HB (2/2)**
>
> > In long sequences, some objects may lead to environmental changes as a result of prior actions. When this occurs, the predictions may lose their overall coherence, necessitating the generation of a new action sequence that takes into account the modified objects.
>
> Certainly earlier actions can change the environment and invalidate future predictions, and MGP-Long is explicitly designed to handle such dynamic and non-Markovian settings. At each step, MGP-Long recomputes the posterior confidence of all future action tokens under the new observation and then edits low-confidence tokens, retaining only high-confidence future tokens that are consistent with the latest observation (Sec 3.4). This dynamic refinement mechanism allows the policy to adapt to environmental changes while preserving global coherence, which is why it excels in dynamic and non-Markovian settings.
>
> Empirically, the results in dynamic environments (Table 4 in the paper) show that when the environment changes or is perturbed, MGP-Long outperforms the MGP-Long w/o sm variant (0.436 vs. 0.396), which discards all high-confidence future tokens and fully regenerates the remaining sequence from scratch. This demonstrates that retaining high-confidence future tokens as anchors is beneficial: they help the model more accurately predict masked tokens and maintain long-horizon consistency (Sec. 4.3 in the paper).
>
> Finally, we note that even MGP-Long w/o sm, which corresponds to always generating a new action sequence conditioned only on the updated environment and executed actions, still significantly outperforms both MGP-full and DP3-full (Table 4 in the paper).
>
>
> We believe this addresses all the points raised – please let us know if you have any further questions.

---

> ### Author Response · Authors · 2025-11-27
> **Additional Reply to reviewer x3HB**
>
> > The results in the simulated environment are impressive. However, due to the sim-to-real gap, it is often necessary to demonstrate effectiveness in real-world settings.
>
> We have now deployed our model operating a LeRobot arm on a real world non-Markovian towel-sorting task. A towel and two baskets are placed in the robot's working area with a random lateral offset. At the beginning of each episode, a light briefly turns on in either red or blue and is then switched off; the color indicates which basket the towel should be placed in. The robot must grasp a towel and place it into the correct basket based solely on the initial light color. As the light is only visible at the very start of the episode, later observations do not contain any information about the desired goal, making the task non-Markovian. We collect 60 expert demonstrations by teleoperation of the LeRobot arm. For each trial, we record robot joint positions, end-effector position and orientation, gripper state, and synchronized RGB, depth, and point-cloud data. We evaluate our model in 25 real-world trials. MGP-Long achieves a success rate of 96%, outperforming DP3-Full Seq (which achieves 84%). Moreover, MGP-Long maintains the same success rate when observations are missing during action execution. These results show that MGP-Long is robust to complex and noisy real-world conditions, and retains its strong global reasoning capabilities in this setting. For more details, see Sec. 5 and Appendix Sec. D in the updated paper.

---

### Author Response · Authors · 2025-11-21
**Common response and updates for revision**

We thank all four reviewers for the positive and detailed feedback. We are particularly grateful for the comments that helped us refine the manuscript by running some important hyperparameter experiments, analyzing and validating the accuracy of confidence scores and assessing the two-stage training complexity and potential domain shift of VQ quantization. We hope we addressed all concerns raised by reviewers, including the following improvements:

- [x3HB, UA2E, 4ZDe, wnrp] **Confidence scores analysis**  We now include additional confidence analyses (confidence heatmaps and mask–unmask plots, and masking-relevance results).
- [x3HB, wnrp] **Real-world evaluation** We are conducting a non-Markovian real-world task (light-cue towel sorting) and additional tests under missing observations, and will report quantitative and qualitative results next week.
- [UA2E] **Multimodality experiments**  We clarify how Gumbel-Max sampling supports multimodality and now include Push-T experiments demonstrating distinct action modes under identical context.
- [UA2E, 4ZDe] **Codebook size & discretization granularity experiments** We provide new ablations on codebook size and discretization granularity, showing stable performance across settings.
- [4ZDe, wnrp] **Two-stage training analysis** We expand our discussion of the two-stage training procedure and include complexity/efficiency analysis and reconstruction checks confirming negligible quantization error for control.
- [x3HB, 4ZDe] **Mask-ratio experiments** We compare multiple mask ratios of MGP-Long on MetaWorld.
- [UA2E] **MGP-Long refinement steps experiments** We evaluate different refinement steps for MGP-Long in dynamic environments.

We provide detailed responses to each reviewer below.

---

### Author Response · Authors · 2025-11-27
**Revised version including real-world evaluation**

We sincerely appreciate the thoughtful feedback from all four reviewers. As suggested, we have now revised the paper to include a real-world evaluation. All updates since the initial version are highlighted in red (including these real-world results but also various additional experiments and analysis). As the discussion period draws to a close, please let us know if you have any further questions or suggestions.

---

### Meta-Review · Area_Chair_rzRE · 2026-01-07

**Summary:**

The recommendation to accept is based on the successful resolution of several key reviewer concerns. Initially, while reviewers praised the novelty and speed of the method, they were hesitant due to a lack of real-world validation, referencing the "sim-to-real" gap common in this domain. Additionally, reviewers questioned the validity of the confidence-based masking mechanism (asking if low confidence actually correlates with failure) and the model's capacity to handle multimodal action distributions given its discrete token nature.

During the rebuttal, the authors provided a compelling response that directly informed this decision. They introduced physical experiments on a LeRobot arm (addressing the sim-to-real concern), provided quantitative analyses and heatmaps validating the confidence scores, and demonstrated multimodal capabilities via Push-T experiments.

**Reviewer Concerns:**

- Real-World Validation: Reviewers x3HB and wnrp strongly requested real-world experiments to bridge the sim-to-real gap. The authors added a real-world non-Markovian "light-cue towel sorting" task using a LeRobot arm, achieving a 96% success rate (vs. 84% for baselines).
- Confidence Score Validity: Reviewers x3HB and UA2E questioned the accuracy and behavior of the confidence scores used for masking. The authors provided heatmaps and quantitative analyses showing that confidence drops align correctly with environmental changes (e.g., dynamic obstacles) and that posterior-based scoring outperforms random scoring.
- Multimodality: Reviewer UA2E asked if the discrete token approach could handle multimodal distributions. The authors clarified their Gumbel-Max sampling strategy and added Push-T experiments demonstrating diverse, valid trajectory samples under identical contexts.
- Hyperparameter Sensitivity & Ablations: Reviewers UA2E and 4ZDe requested ablations on codebook size, mask ratios, and discretization granularity. The authors performed these extensive ablations, showing the method is robust to codebook size (512 vs 2048) and token granularity (2 vs 8 actions/token), and identified optimal mask ratios (70%).

- Reviewer 4ZDe noted that "joint end-to-end training" (Transformer + VQ-VAE) might be better than the current two-stage pipeline. The authors acknowledged this as a valid future direction but justified the current standard two-stage approach using references to similar state-of-the-art works (e.g., MaskGIT, VQ-BeT). This is a suggestion for future work rather than a flaw in the current submission.

**Reviewer Scores:**

Reviewers had all given initial positive reviews and the rebuttal has sufficiently addressed the concerns and I think the reviewers would've maintained their positive reviews.

---

### Decision · Program_Chairs · 2026-01-26

Accept (Poster)